

# Froissart bound for/from CFT Mellin amplitudes

**Parthiv Haldar[1⋆] and Aninda Sinha[1†]**

**1** Centre for High Energy Physics, Indian Institute of Science,
C.V. Raman Avenue, Bangalore 560012, India.

⋆ parthivh@iisc.ac.in, † asinha@iisc.ac.in

## Abstract

We derive bounds analogous to the Froissart bound for the absorptive part of CFT$_d$ Mellin amplitudes. Invoking the AdS/CFT correspondence, these amplitudes correspond to scattering in AdS$_{d+1}$. We can take a flat space limit of the corresponding bound. We find the standard Froissart-Martin bound, including the coefficient in front for $d + 1 = 4$ being $\pi/\mu^2$, $\mu$ being the mass of the lightest exchange. For $d > 4$, the form is different. We show that while for $CFT_{d\leq6}$, the number of subtractions needed to write a dispersion relation for the Mellin amplitude is equal to 2, for $CFT_{d>6}$ the number of subtractions needed is greater than 2 and goes to infinity as $d$ goes to infinity.



# 1 Introduction

The famous Froissart bound [1–3], for total scattering cross-section, states that in the forward limit, the high energy behaviour is bounded by

$$\sigma_{tot} < \frac{\pi}{\mu^2} \log^2 \frac{S}{S_0}, \tag{1.1}$$

where $\mu$ is the mass of the lightest exchanged particle in $T-$channel, $S$ is the usual Mandelstam variable and $S_0$ is an constant having the dimensions of $S$. The main assumptions that go into deriving this are a) Unitarity b) Analyticity c) Polynomial boundedness. The scattering process being described is 2-2 scattering involving identical *massive* hadrons with *no* massless exchanges. Typically $\mu$ is the mass of the pion. There is a lot of interest [4,5] to know how close experimental data, in the high energy limit, is to saturating this bound. From experimental fits of proton-proton data, the coefficient $\pi/\mu^2$ works out to be around two orders of magnitude bigger than what data suggests–thus, unlike what theory suggests, if $\mu$ was the mass of the external particle, agreement would be better. There have been attempts to figure out how to

make the bound stronger [6,7] but with virtually no success so far[1]. Analogous results for the absorptive part of the scattering amplitude can be worked out in any spacetime dimensions [9,10]. The absorptive part is related to the total cross section via the optical theorem.

In this paper, we will initiate the examination of the Froissart bound using conformal field theory techniques. The idea is roughly as follows. There exists a way of representing the four point correlation function of conformal primary operators in Mellin space. The Mellin variables $s, t$ are the analogues of Mandelstam variables $S, T$. Mellin techniques for CFTs have been developed during the last 10 years after the pioneering work of Mack [11–14]. The Mellin amplitude can be thought of as a representation of scattering in anti-de Sitter space. The CFT lives in $d$ spacetime dimensions while the scattering happens in $d + 1$ dimensional AdS space. Note that if we want to talk about the Froissart bound in 4 spacetime dimensions then we need to examine the Mellin amplitude in $CFT_{d=3}$. If the radius of curvature of the AdS space $R$ is much bigger than the Planck length $\ell_P$, then effectively the cosmological constant is zero and the scattering can be thought to be occurring in flat spacetime–see fig.1. Precise formulae have been conjectured [15] (with a perturbative proof) for the relation between the Mellin amplitude and flat space scattering of massive particles. The salient feature to remember for now is that the AdS/CFT dictionary gives $mR \sim \Delta_\phi$, where $m$ is the mass of the external particle (scalar for our discussion) and $\Delta_\phi$ is the dimension of the CFT conformal primary dual to this scalar. Here, we assume $m$ fixed while $R/\ell_P \gg 1$ as well as $mR \sim \Delta_\phi \gg 1$. We will give the precise map later on.

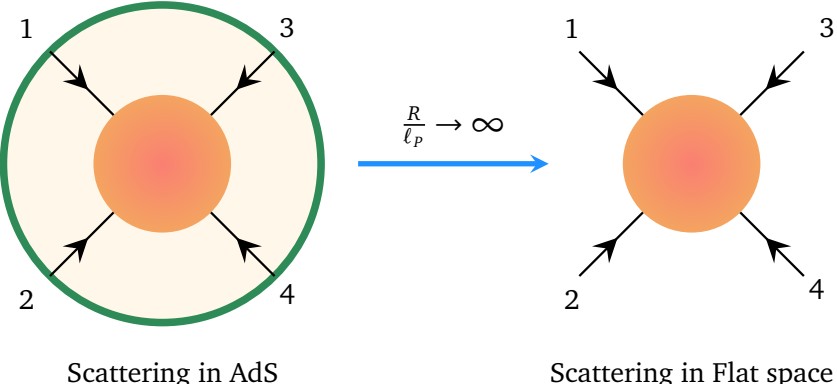

Figure 1: Transition from AdS to Flat Space

Now, that such a dictionary exists, it is natural to ask what is the analog of the Froissart bound for Mellin amplitude and then via this dictionary, what happens to the flat space limit[2]. Namely, can we get a different coefficient in front of the bound in eq.(1.1)? In the future, one can also hope to compute subleading $1/R$ corrections to this bound, which we will sometimes refer to as the Froissart$_{AdS}$ bound to distinguish from the flat space Froissart bound. Let us summarize the methodology we will adopt:

1. As in the Froissart bound derivation, we start with the absorptive (imaginary) part of the amplitude. However, unlike in flat space where there is a cut in the complex $S$ plane, in the Mellin variable $s$, we have an infinite set of poles in the Mellin amplitude. In the imaginary part, these poles become a sum of delta functions [16].

2. The flat space amplitude is expanded in terms of the Gegenbauer polynomials which are the generalizations of the Legendre polynomials. The polynomials are indexed by

---

[1]See [8] for a recent discussion.

[2]Hence the for/from in our title!

the spin quantum number $\ell$. The Mellin amplitude is expanded in terms of the so-called Mack polynomials which are indexed by a spin quantum number $\ell$, as well as the dimension $\Delta$ of the exchanged conformal primary. In the flat space limit, to be described below, however, the Mack polynomials go over to the Gegenbauer polynomials.

3. In the flat space derivation, an assumption is made about the polynomial boundedness of the amplitude inside the Martin ellipse. We will make a similar assumption about the Mellin amplitude[3]. This assumption effectively leads to the sum over $\ell$ being cut-off. Typically the cut-off $L$ takes the form

$$L \propto \sqrt{S} \ln \frac{S}{S_0},$$

with the proportionality constant depending on the power assumed in the polynomial boundedness.

4. The key difference will be that unlike the partial wave unitarity bounds that are assumed in the flat space derivation, we will have to contend with the sum over $\Delta$. Here, we will make use of the fact that in order to reproduce the identity exchange in the crossed channel, the operator product expansion coefficients governing the sum over $\Delta$ is controlled by the so-called (complex) Tauberian theorems [17,18]. A final point to mention is that we will be dealing with averaged bounds, following for example [19]. This is because we will be dealing with distributions and it makes more sense to talk about integrated quantities. This will turn out to be essential in making the range of twists in the $\Delta$ sum on the CFT side to be finite.

The last point creates a very important difference in the form of the Froissart$_{AdS}$ bounds we will find. Summarizing our results:

- We find that the coefficient in front of the bound, i.e., $\pi/\mu^2$ is *exactly* this for 4 flat spacetime dimensions *except* that $\mu$ here can also be the mass of the external particle while the mass parameter present in the original Froissart bound formula eq.(1.1) is the mass of the lightest exchanged particle in $T$ channel, usually taken to be the pion mass.

- For $CFT_d$ with $d \leq 4$, the form of the bound is the same as flat space higher dimensional generalizations. However, for $d = 2$ the coefficient in front is *lower* than the flat space derivation, for $d = 3$ it is *identical* as mentioned above, while for $d = 4$ it is *bigger*.

- For $d > 4$ the form of the bound is different, as we will discuss in our derivation below. This has important implications for the form of the polynomial boundedness. What happens is that first one assumes that the amplitude is $|\mathcal{M}(s,t)| < s^n$ bounded for some unspecified $n$ for $t$ inside the Martin ellipse. Then, this leads to the Froissart bound (the $n$ enters in the coefficient in front). Suppose that the result for the absorptive part is $\mathcal{A}_M(s,0) < cs^a \ln^{d-1} s/s_0$. At this stage, one can argue using the Phragmen-Lindeloff theorem [20] that $n \leq \lfloor a \rfloor + 1$, where $\lfloor \ \rfloor$ denotes the usual floor function. In the flat space derivation $\lfloor a \rfloor = 1$. However, we will find $\lfloor a \rfloor > 1$, and hence $n > 2$ for $d > 6$.

We will attempt to keep this paper self-contained and hence will review several scattered results wherever necessary. The paper is organized as follows. We begin by reviewing old literature concering flat space Froissart bound in section 2. In section 3, we review Mellin amplitudes in CFTs including the flat space limit reviewing essential results from [15]. In section 4, we turn to bounding the absorptive part of the Mellin amplitude. We derive dispersion relations in section 5 and constrain the number of subtractions needed. We conclude in section

---

[3]We made explicit checks using the mean field theory OPE coefficients for the validity of this assumption.

6. There are appendices supplementing the calculations in the main text.

**Note:** Our approach and findings are complementary to the recent paper [21] where the techniques rely on the absence of certain spurious singularities [22] and do not admit an obvious flat space limit leading to the Froissart bound.

*Warning: We will use the convention $h = d/2$ in many places. This unfortunate convention is somewhat standard in the CFT literature.*

## 2 The flat space story: A brief review

In this section, we would like to review some standard results for flat space scattering amplitude theory. In the early days of axiomatic quantum field theory, various analytical statements about quantum field theory were proved [1, 2, 19, 23–29] in general without any recourse to perturbative methods[4]. The common feature, that all these proofs shared, is that only basic requirement of **unitarity** and elementary assumptions about **analytic structure** of the scattering amplitude were used.

### 2.1 Kinematics

We start with reviewing basic kinematical structure of a $2 \to 2$ scattering amplitude involving identical massive scalar particles with mass $m$ in Minkowski space-time $\mathbb{M}^{d,1}$. The scattering configuration is as in 1. We will focus our attention upon the corresponding scattering amplitude $\mathcal{T}(\{p_i^\mu\})$. The Mandelstam variables $(S, T, U)$ are defined as,

$$S = -(p_1^\mu + p_2^\mu)^2, \quad T = -(p_1^\mu + p_4^\mu)^2, \quad U = -(p_1^\mu + p_3^\mu)^2. \tag{2.1}$$

Here $\{p_i^\mu\}$ are the Minkowski-momenta of the scattering particles constrained by conservation,

$$\sum_i p_i^\mu = 0. \tag{2.2}$$

Also the Mandelstam variables satisfy the usual constraint,

$$S + T + U = 4m^2. \tag{2.3}$$

### 2.2 Partial wave expansion

Now we turn to the dynamical consideration of the scattering amplitude. At the centre stage of the **rigorous unitarity-analyticity** program for scattering amplitude is the partial wave expansion of the scattering amplitude. The $d + 1$ dimensional flat space scattering amplitude admits a partial wave expansion in terms of generalized spherical functions spanning the representation space of $SO(d, 1)$ corresponding to the unitary irreducible representations of maximally compact subgroup $SO(d)$. The $2 \to 2$ scattering amplitude $\mathcal{T}(S, T)$ admits the following $S-$channel partial wave expansion [9] in a basis of Gegenbauer polynomials,

$$\mathcal{T}(S, T) = \phi(s) \sum_{\substack{\ell=0 \\ \ell \text{ even}}}^{\infty} f_\ell(S) \frac{C_\ell^{(h-1)}(1)}{N_\ell^{(h-1)}} C_\ell^{(h-1)}\left(Z_S = 1 + \frac{2T}{S - 4m^2}\right), \tag{2.4}$$

---

[4]See also [3].

with $C_\ell^{(h-1)}$ being the Gegnbauer polynomial and $\{f_\ell(S)\}$ are the *partial wave coefficients*. Here,

$$
\begin{aligned}
h &= \frac{d}{2}, \\
N_\ell^{(h-1)} &= \frac{2^{3-2h}\pi\Gamma(\ell+2h-2)}{\ell!(h-1+\ell)\Gamma^2(h-1)}, \\
\phi(s) &= 2\Gamma\left(h-\frac{1}{2}\right)(16\pi)^{\frac{2h-1}{2}} s^{\frac{3-2h}{2}}, \\
C_\ell^{(h-1)}(1) &= \frac{(2h-2)_\ell}{\Gamma(\ell+1)},
\end{aligned}
\tag{2.5}
$$

where $(a)_b$ denotes the Pochhammer symbol.

## 2.3 Implication of unitarity: Partial wave bound

Unitarity implies boundedness of the partial wave coefficients. More specifically:

$$
0 \le |f_\ell(S)|^2 \le \mathrm{Im}[f_\ell(S)] \le 1 \,.
\tag{2.6}
$$

In particular, this has the important implication that $\mathrm{Im}[f_\ell(S)]$ is *positive* and bounded above by unity. This implication is often dubbed as **positivity** and this particular piece of result plays a crucial role in the unitarity-analyticity program. In this program, the quantity $\mathrm{Im}[\mathcal{T}(S,T)]$ plays a very important role. In fact, while proving the Froissart-Martin bound it is this quantity which is bounded and then the forward scattering cross-section is bounded by its relation to the former via optical theorem. $A_S(S,T) \equiv \mathrm{Im}[\mathcal{T}(S,T)] := \lim_{\epsilon\to 0}[\mathcal{T}(S+i\epsilon,T)-\mathcal{T}(S-i\epsilon,T)]/2i$ is also called **absorptive part** of the scattering amplitude.

## 2.4 Implication of analyticity

Now we turn to the main analytitcity properties of the scattering amplitude $\mathcal{T}(S,T)$ that follows from the local field theory. For this we will interchangebly use $\mathcal{T}(S,T)$ and $\mathcal{T}(S,Z_S)$. $Z_S$ was defined in eq.(2.4). Lehmann [24] showed, starting from the principles of the local field theory, that $\mathcal{T}(S,Z_S)$ is analytic in $Z_S$ in an ellipse with foci in $Z_S = \pm 1$. This ellipse is called **Lehmann ellipse**. $\mathrm{Im}[\mathcal{T}(S,Z_S)]$ is analytic in a larger ellipse, the "large" Lehmann ellipse. Martin [27,28] enlarged the ellipses and proved that for fixed $S$ near a physical point, $\mathcal{T}(S,T)$ is analytic in $|T| < R$ where $R$ is *independent* of $S$. This result also holds for $\mathrm{Im}[\mathcal{T}(S,Z_S)]$. For our purpose $R = 4m^2$.

## 2.5 Polynomial boundedness

Polynomial boundedness is a very crucial ingredient that goes into derivation of the Froissart-Martin bound. According to it [20], there exists a certain finite number $R$ and a positive integer $N$ such that one has,

$$
A_S(S, T = R) < cS^N \,.
\tag{2.7}
$$

More rigorously, this condition is expressed by the convergence of the integral,

$$
\int_{4m^2}^{\infty} \frac{dS'}{S'^{N+1}} A_S(S', T = R) \,.
\tag{2.8}
$$

## 2.6 Froissart-Martin bound

We provide a derivation for $3 + 1$ dimensional Minkowski spacetime in appendix C. In the work in the following sections, we will follow closely the steps of that proof. Using the partial wave unitarity bound, Martin analyticity, and polynomial boundedness, one can derive the following asymptotic bound on the absorptive part of the scattering amplitude, $A_S(S, T = 0)$, for $S \to \infty$ [9],

$$A_S(S, T = 0) \leq 2^{4h-3} \pi^{h-2} \frac{\Gamma(h-1)\Gamma^2(h-1)}{(2h-1)\Gamma^2(2h-2)} \left( \frac{N-1}{\sqrt{R} \cos \varphi_0} \right)^{2h-1} S(\ln S)^{2h-1}, \tag{2.9}$$

where $R$ is same as in eq.(2.7). In $\mathbb{M}^{3,1}$ i.e., $d = 3$, one can further obtain from this, via optical theorem, the famous Froissart-Martin bound on high energy total scattering cross-section in forward limit,

$$\sigma_{tot} \leq \frac{\pi}{\mu^2} \ln^2 \frac{S}{S_0}, \tag{2.10}$$

where $S_0$ is a constant having the dimesnion of $S$ and $\mu$ is the mass of the lightest exchanged particle in crossed channel. One needs to put $R = 4\mu^2$, $N = 2$ and $\cos \varphi_0 = 1$ into eq.(2.9) to obtain this bound. That $N = 2$ is required was proved in [26] which basically implies that *it is possible to write a fixed T dispersion formula for scattering amplitude with atmost two subtractions*. The value of $R$ is dictated by the Martin analyticity.

# 3 Mellin amplitude in CFTs

## 3.1 Definitions and conventions

Mellin amplitudes for CFT correlators were introduced by Mack [11, 12]. In this section, we will review the analogy between conformal correlation function and scattering amplitude [13, 14, 30] via the AdS/CFT correspondence. In particular, one can consider the Mellin amplitude as the "scattering amplitude in AdS".

The Mellin amplitude associated with connected part of the $n$-point function of scalar primary operators is defined by,

$$G(x_i) = \langle \mathcal{O}_1(x_1) \ldots \mathcal{O}_n(x_n) \rangle_c = \int [d\delta] \mathcal{M}(\delta_{ij}) \prod_{1 \leq i < j \leq n} \frac{\Gamma(\delta_{ij})}{(x_{ij}^2)^{\delta_{ij}}}, \tag{3.1}$$

where the integral runs parallel to the imaginary axis and is to be understood in the sense of Mellin-Barnes contour integral. Conformal invariance constrains the integration variables $\{\delta_{ij}\}$ to satisfy,

$$\delta_{ij} = \delta_{ji}, \quad \delta_{ii} = -\Delta_i, \quad \sum_{j=1}^{n} \delta_{ij} = 0, \tag{3.2}$$

with $\Delta_i$ being the scaling dimension of the operator $\mathcal{O}_i$. Due to these constraints, there are $n(n-3)/2$ independent variables upon which the Mellin amplitude $\mathcal{M}$ depends. Clearly, for four-point function i.e., $n = 4$, the number of independent variables is 2.

Now let us focus on the problem at hand for which we will consider $4$−point correlator of identical scalar primaries $\phi$ with dimension $\Delta_\phi$. The reduced correlator $\mathcal{G}(u, v)$ is defined by,

$$\langle \phi(x_1)\phi(x_2)\phi(x_3)\phi(x_4) \rangle = \frac{1}{x_{12}^{2\Delta_\phi} x_{34}^{2\Delta_\phi}} \mathcal{G}(u, v), \tag{3.3}$$

where $u, v$ are the conformal cross-ratios given by $u = \frac{x_{12}^2 x_{34}^2}{x_{13}^2 x_{24}^2}$, $v = \frac{x_{14}^2 x_{23}^2}{x_{13}^2 x_{24}^2}$. Now we define the "Mellin variables" $(s, t)$ as:

$$
\begin{aligned}
\delta_{12} = \delta_{34} &= \frac{\Delta_\phi}{2} - s, \\
\delta_{14} = \delta_{23} &= \frac{\Delta_\phi}{2} - t, \\
\delta_{13} = \delta_{24} &= s + t.
\end{aligned}
\tag{3.4}
$$

Note that this definition differs from the ones in [31] by a shift of $\Delta_\phi/2$. The reduced correlator now has the Mellin space represenatation,

$$
\mathcal{G}(u, v) = \int_{-i\infty}^{i\infty} \frac{ds}{2\pi i} \frac{dt}{2\pi i} u^{s + \Delta_\phi/2} v^{t - \Delta_\phi/2} \mu(s, t) \mathcal{M}(s, t),
\tag{3.5}
$$

with

$$
\mu(s, t) = \Gamma^2\left(\frac{\Delta_\phi}{2} - s\right) \Gamma^2\left(\frac{\Delta_\phi}{2} - t\right) \Gamma^2(s + t),
\tag{3.6}
$$

being a standard measure factor which has information about the double trace operators in the $N \to \infty$ limit in the context of the AdS/CFT correspondence.

## 3.2 Conformal Partial Wave expansion

Just as the flat space scattering amplitude admits a partial wave expansion in terms of Gegenbauer polynomials, the Mellin amplitude $\mathcal{M}(s, t)$ admits the conformal partial wave expansion [11, 32]. Our starting point is the Mellin space representation of the standard position space direct channel expansion [32]. We are interested in the imaginary part of this which arises from the physical poles. We can either work directly with [32] or a bit more conveniently, to make the pole structure manifest, following [31], we can write an $s-$channel conformal partial wave expansion for the Mellin amplitude,

$$
\mathcal{M}(s, t) = \sum_{\substack{\tau, \ell \\ \ell \text{ even}}} C_{\tau, \ell}\, f_{\tau, \ell}(s)\, \widehat{\mathcal{P}}_{\tau, \ell}(s, t),
\tag{3.7}
$$

where the equality is modulo some *regular terms* –see appendix D for a derivation of how to go from the form in [32] to the form in [31]. Here, we have defined $\tau = \frac{\Delta - \ell}{2}$ and $\widehat{\mathcal{P}}_{\tau, \ell}(s, t)$ are the Mack polynomials whose details we provide in appendix A. $C_{\tau, \ell}$ are the squared OPE coefficients and

$$
\begin{aligned}
f_{\tau, \ell}(s) = &\frac{\mathcal{N}_{\tau, \ell}\, \Gamma^2(\tau + \ell + \Delta_\phi - h)}{\left(\tau - s - \frac{\Delta_\phi}{2}\right) \Gamma(2\tau + \ell - h + 1)} \frac{\sin^2 \pi\left(\frac{\Delta_\phi}{2} - s\right)}{\sin^2 \pi\left(\Delta_\phi - \tau - \frac{\ell}{2}\right)} \\
&\times\; {}_3F_2\left[\begin{matrix} \tau - s - \frac{\Delta_\phi}{2}, 1 + \tau - \Delta_\phi, 1 + \tau - \Delta_\phi \\ 1 + \tau - s - \frac{\Delta_\phi}{2}, 2\tau + \ell - h + 1 \end{matrix} \middle| 1\right],
\end{aligned}
\tag{3.8}
$$

$$
\tag{3.9}
$$

with,

$$
\mathcal{N}_{\tau, \ell} := 2^\ell \frac{(2\tau + 2\ell - 1)\Gamma^2(2\tau + 2\ell - 1)\Gamma(2\tau + \ell - h + 1)}{\Gamma(2\tau + \ell - 1)\Gamma^4(\tau + \ell)\Gamma^2(\Delta_\phi - \tau)\Gamma^2(\Delta_\phi - h + \tau + \ell)}.
\tag{3.10}
$$

${}_3F_2$ is a generalized hypergeometric function. There are poles at $s = \tau - \frac{\Delta_\phi}{2} + q$ for $q \in \mathbb{Z} \geq 0$. This representation is suitable for the $s$ channel Witten diagram and the residues at the physical

poles are identical to other standard ones used in the literature eg. [32]. Note that the *full* Mellin amplitude includes the measure factor, which provides the *u*-channel poles. However, in what follows, we will be interested in averaging over positive values of *s* so that these poles will not alter any of our conclusions–this is analogous to the flat space derivation in [19] and is reviewed in appendix C.

## 3.3 Flat space limit of the Mellin amplitude

Now we will review the connection between the Mellin amplitude and the flat space scattering via what is called the "flat space limit". That such a connection exists was first conjectured in [13] and was developed extensively in [16, 33]. In these papers, the Mellin amplitude was related to scattering amplitude of *massless* particles. However, the limit in which we are interested is the so called "massive flat space limit" and was first proposed in [15]. In this limit, the Mellin amplitude for the conformal correlator is related to the scattering amplitude for massive particles by taking the dimensions of the external operators to be parametrically large. Then via the AdS/CFT correspondence, the Mellin amplitude of conformal correlator is related to flat space scattering amplitude with external *massive* particles in one higher space-time dimesnion i.e., the Mellin amplitude of a conformal correlator in $CFT_d$ is related to scattering amplitude $d + 1$ dimensional flat space quantum field theory–to emphasise, the flat space QFT is not conformal.

To understand better what we mean by parametrically large dimension, recall that in the AdS/CFT correspondence the scaling dimension of the boundary conformal operators in $CFT_d$, $\Delta_\phi$, and the mass of the corresponding dual bulk field in $AdS_{d+1}$, $m$, are related by

$$m^2 R^2 = \Delta_\phi (\Delta_\phi - d), \tag{3.11}$$

with $R$ being the AdS radius. Now in the flat space limit the conformal dimension $\Delta_\phi$ is taken to infinity along with $R$ so that $m$ remains finite i.e.,

$$\lim_{\substack{\Delta_\phi \to \infty \\ R \to \infty}} \frac{\Delta_\phi^2}{R^2} = m^2. \tag{3.12}$$

Since $R$ is dimensionful, we mean $R/\ell_P \gg 1$ where $\ell_P$ is the Planck length. Further, since we are taking the flat space limit to a massive theory, we also require $R/\ell_s$ with $\ell_s$ being the string length characterizing the string theory energy scale. Now taking $R/\ell_P \to \infty$ takes the $AdS_{d+1}$ to $\mathbb{M}^{d,1}$. Thus, in this limit, we relate the Mellin amplitude for $CFT_d$ correlator to scattering amplitude of massive particles in flat spacetime $\mathbb{M}^{d,1}$. Now we turn to the explicit formulae relating the flat spacetime scattering amplitude and the Mellin amplitude.

The $n$−point conformal correlator and $n$−particle scattering amplitude are related by:

$$(m_1)^a \mathcal{T}\left(\{p_i^\nu\}\right) = \lim_{\substack{\Delta_i \to \infty \\ R \to \infty}} \frac{(\Delta_1)^a}{\mathcal{N}} \mathcal{M}\left(\delta_{ij} = \frac{\Delta_i \Delta_j + R^2 \, p_i^\nu p_{j\nu}}{\Delta_1 + \cdots + \Delta_n} + O\left(\Delta_1^0\right)\right), \tag{3.13}$$

with

$$\mathcal{N} := \frac{1}{2} \pi^{\frac{d}{2}} \Gamma\left(\frac{\sum \Delta_i - d}{2}\right) \prod_{i=1}^n \frac{\sqrt{\mathcal{C}_{\Delta_i}}}{\Gamma(\Delta_i)}, \qquad \mathcal{C}_\Delta := \frac{\Gamma(\Delta)}{2\pi^{\frac{d}{2}} \Gamma\left(\Delta - \frac{d}{2} + 1\right)}, \qquad a := \frac{n(d-1)}{2} - d - 1. \tag{3.14}$$

Here $\mathcal{T}\left(\{p_i^\nu\}\right)$ is the $n$-particle $\mathbb{M}^{d,1}$ scattering amplitude with external Minkowski momenta $\{p_i^\nu\}$. Note however, these $\{p_i^\nu\}$ have momenta interpretation after going to flat space amplitude only. On the Mellin amplitude side they are just $n$ vectors in $\mathbb{M}^{d,1}$ with the restriction,

$$\sum_{i=1}^{n} p_i^{\gamma} = 0, \qquad p_i^{\gamma} p_{i\nu} = -\frac{\Delta_i(\Delta_i - d)}{R^2}. \tag{3.15}$$

These restrictions are there for consistency with the momentum interpretation of $\{p_i^{\gamma}\}$ in the flat space limit. Note that the vector norm and inner product are usual $\mathbb{M}^{d,1}$ norm and inner product respectively. The parameterization holds for $\delta_{ij}$ with $i \neq j$. $\delta_{ii}$ should still be set to $-\Delta_i$ explicitly. Now the consistency with the third constraint in eq.(3.2) can be met by adding following finite term

$$\frac{d}{n-2}\left[\frac{\Delta_i + \Delta_j}{\Delta_1 + \cdots + \Delta_n} - \frac{1}{n-1}\right]. \tag{3.16}$$

[15] gave a perturbative proof for eq.(3.13).

For the case of the 4-point conformal correlator of identical scalar primaries $\phi$ with scaling dimensions $\Delta_{\phi}$ and the corresponding flat space mass $m$ we have

$$\boxed{m^a \, \mathcal{T}(S, T) = \lim_{\Delta_{\phi} \to \infty} \frac{(\Delta_{\phi})^a}{\mathcal{N}} \, \mathcal{M}(s, t),} \tag{3.17}$$

with

$$\mathcal{N} := \frac{\Gamma(2\Delta_{\phi} - h)}{8\pi^h \Gamma^2(\Delta_{\phi})\Gamma^2(\Delta_{\phi} - h + 1)}, \qquad a := 2h - 3. \tag{3.18}$$

Here $(s, t)$ are defined as in eq.(3.4) and the flat space Mandelstam variables $(S, T)$ are those defined in eq.(2.1). From the precise relation between the flat space Mandelstam variables $(S, T)$ and $(s, t)$ is given by:

$$s(t) = \frac{R^2}{8\Delta_{\phi}} S(T) + \frac{d}{6}. \tag{3.19}$$

The term $d/6$ is the finite term eq.(3.16) which we can ignore for all practical purpose in the flat space limit and thus we are going to use for all practical purposes,

$$s(t) = \frac{R^2}{8\Delta_{\phi}} S(T). \tag{3.20}$$

On the same footing we use

$$\hat{u} = \frac{R^2}{8\Delta_{\phi}} U \tag{3.21}$$

so that we can consider,

$$s + t + \hat{u} = \frac{\Delta_{\phi}}{2}. \tag{3.22}$$

Note the consistency of the constraints eq.(3.22) and eq.(2.2). Then drawing parallels with the flat space understanding, the "physical domain" for $s$−channel of the Mellin amplitude in the current perspective is defined to be

$$s > \Delta_{\phi}/2; \quad t, \hat{u} < 0. \tag{3.23}$$

Since the flat space limit is $R \to \infty$, $R$ being the AdS radius, we would like to have a $1/R$ expansion around the flat space. On dimensional grounds, in terms of the flat space $S$, we expect the dimensionless quantities $S/(m^4 R^2)$ and $1/(SR^2)$ to be small in order to allow a $1/R$ expansion. Here we are assuming only even powers of $m$ entering such expansion. In terms of $s$, this gives $(2\Delta_{\phi})^3 \gg s \gg 2\Delta_{\phi} \gg 1$ which is what is going to be used below.

### 3.4 "Absorptive Part" of Mellin amplitude

The main goal in this work is to extract information about Mellin amplitude for conformal field theory by exploiting the structural analogy between the former and the flat space scattering amplitude. The absorptive part of scattering amplitude is the imaginary part of scattering amplitude. In this spirit, we define the **absorptive part of the Mellin amplitude** as,

$$\mathcal{A}_M(s,t) = \mathrm{Im}_s\,\mathcal{M}(s,t) = \sum_{\substack{\tau,\ell \\ \ell\ \mathrm{even}}} C_{\tau,\ell}\,\mathrm{Im}_s[f_{\tau,\ell}(s)]\,\widehat{\mathcal{P}}_{\tau,\ell}(s,t), \tag{3.24}$$

where we have defined,

$$\mathrm{Im}_s[g(s)] := \lim_{\varepsilon\to 0}\frac{g(s+i\varepsilon)-g(s-i\varepsilon)}{2i}. \tag{3.25}$$

Observe that the imaginary part comes only from the fucntion $f_{\tau,\ell}$ because for unitary theories $C_{\tau,\ell}\in\mathbb{R}^+$ and $\widehat{\mathcal{P}}_{\tau,\ell}(s,t)$ does not have poles. The imaginary part of the function $f_{\tau,\ell}$ comes in a distributional sense at the pole locations which we will see in a while. Since $\mathcal{A}_M(s,t)$ is a distribution, we should handle quanitites involving integrals over $\mathcal{A}_M(s,t)$. Towards that end, we define the following quantity [19]

$$\bar{\mathcal{A}}_M(s,t) \equiv \frac{1}{s-\frac{\Delta_\phi}{2}}\int_{\frac{\Delta_\phi}{2}}^{s} ds'\,\mathcal{A}_M(s',t). \tag{3.26}$$

This can be viewed as an averaged absorptive Mellin amplitude. For $d=3$, which leads to Froissart bounds for 4d flat space, the choice of the lower limit makes no difference. We will introduce the quantity $x = 1 + 2t/(s-\Delta_\phi/2)$. We will consider the problem of obtaining the asymptotic upper bound on this quantity in the limit $s\to\infty$ for two different scenarios: one is the "forward" limit i.e., $x\to 1$ and the other one is the "non-forward" limit i.e., with $x\neq 1$.

### 3.5 Polynomial boundedness of Mellin amplitude

Now to proceed further, we need to assume something more about the analytic structure of the Mellin amplitude. Recall that the assumption of polynomial boundedness of the flat space scattering amplitude is extremely crucial in deriving the Froissart-Martin bound. In fact, it will be no exaggeration to say that the Froissart-Martin bound would not have existed without this additional boundedness property of the scattering amplitude. We will assume a similar polynomial boundedness for Mellin amplitudes as well. In close analogy with flat space case we assume the following polynomial boundedness condition upon $\mathcal{A}_M(s,t)$: there exists at least an $n\in\mathbb{Z}^+$ such that the integral,

$$\mathfrak{a}_{n,\rho} := \int_{\frac{\Delta_\phi}{2}}^{\infty}\frac{d\bar{s}}{\bar{s}^{n+1}}\mathcal{A}_M(\bar{s},\rho\frac{\Delta_\phi}{2}) \tag{3.27}$$

exists. For our purpose we can assume $\rho\in\mathbb{R}^+$. In the flat space limit, this corresponds to $T = 4\rho m^2$ with $m$ being the mass of the external particle. In the flat space Froissart bound, one typically chooses $\rho = \mu^2/m^2$ with $\mu\le m$ being the mass of the lightest exchange in the crossed channel.

### 3.6 The structure of $\mathcal{A}_M(s,t)$

We will now bound the quantity $\bar{\mathcal{A}}_M(s)$. To do so, we will need to know the structure of $\mathcal{A}_M(s,t)$. The most non-trivial component of the same is $\text{Im}[f_{\tau,\ell}]$. From eq.(3.8) one has ,

$$
\begin{aligned}
{}_3F_2 & \left[ \begin{matrix} \tau - s - \frac{\Delta_\phi}{2}, 1+\tau-\Delta_\phi, 1+\tau-\Delta_\phi \\ 1+\tau-s-\frac{\Delta_\phi}{2}, 2\tau+\ell-h+1 \end{matrix} \, \middle| \, 1 \right] \\
& = \sum_{q=0}^{\infty} \frac{(1+\tau-\Delta_\phi)_q^2}{q!(2\tau+\ell-h+1)_q} \frac{\tau-s-\Delta_\phi/2}{q+\tau-s-\Delta_\phi/2} .
\end{aligned}
\tag{3.28}
$$

Clearly, we see that we have poles in the $s-$plane at the locations $s = \tau - \Delta_\phi/2 + q$ for $q \geq 0$. Now we know that at the poles the imaginary part comes as a Dirac-delta distribution i.e.,

$$
\text{Im.} \frac{1}{x-a} = \lim_{\varepsilon \to 0} \frac{1}{2i} \left( \frac{1}{x-a+i\varepsilon} - \frac{1}{x-a-i\varepsilon} \right) = -\pi\delta(x-a).
\tag{3.29}
$$

Note that this is a distributional statement and hence this equality "holds under the integrals". Specifically if $f(x)$ be a Schwartz function over $\mathbb{R}$ then we have,

$$
\int_{-\infty}^{\infty} dx \, f(x) \, \text{Im.} \left( \frac{1}{x-a} \right) = -\pi f(a).
\tag{3.30}
$$

So in this sense we can write,

$$
\begin{aligned}
& \frac{\text{Im.} \, {}_3F_2 \left[ \begin{matrix} \tau-s-\frac{\Delta_\phi}{2}, 1+\tau-\Delta_\phi, 1+\tau-\Delta_\phi \\ 1+\tau-s-\frac{\Delta_\phi}{2}, 2\tau+\ell-h+1 \end{matrix} \, \middle| \, 1 \right]}{\left( s + \frac{\Delta_\phi}{2} - \tau \right)} = \\
& -\pi \sum_{q=0}^{\infty} \frac{(1+\tau-\Delta_\phi)_q^2}{q!(2\tau+\ell-h+1)_q} \delta(-q-\tau+s+\Delta_\phi/2)
\end{aligned}
\tag{3.31}
$$

Thus collecting everything together we have,

$$
\begin{aligned}
\text{Im.}[f_{\tau,\ell}(s)] = & \pi \mathcal{N}_{\tau,\ell} \frac{\Gamma^2(\tau+\ell+\Delta_\phi-h)}{\Gamma(2\tau+\ell-h+1)} \frac{\sin^2 \pi \left( \frac{\Delta_\phi}{2} - s \right)}{\sin^2 \pi \left( \Delta_\phi - \tau - \frac{\ell}{2} \right)} \\
& \times \sum_{q=0}^{\infty} \frac{(1+\tau-\Delta_\phi)_q^2}{q!(2\tau+\ell-h+1)_q} \delta(-q-\tau+s+\Delta_\phi/2).
\end{aligned}
\tag{3.32}
$$

We would like to note further that for $q = 0$ corresponds to the contributions from the primary while the $q \neq 0$ corresponds to that coming from the descendants. In appendix G, we consider bounds on the primary contribution separately. This exercise is instructive, although the bounds thus obtained are exponentially smaller for large $\Delta_\phi$ compared to the full consideration in the next section.

## 4 Bounds

### 4.1 Obtaining the Froissart$_{AdS}$ bound: "Forward Limit"

We start with the expression for the conformal partial wave expansion of the Mellin amplitude as defined in eq.(3.7). Further making use of eq.(3.28), we can write the meromorphic

structure of the Mellin amplitude as following,

$$
\mathcal{M}(s,t) = -\sum_{\tau,\ell} C_{\tau,\ell}\, \mathcal{N}_{\tau,\ell}\,\Gamma(2\Delta_\phi + \ell - h)\,\frac{\sin^2 \pi\left[\frac{\Delta_\phi}{2} - s\right]}{\sin^2 \pi\left[\Delta_\phi - \tau - \frac{\ell}{2}\right]}\,\widehat{\mathcal{P}}_{\tau,\ell}(s,t)\left(\sum_{q=0}^{\infty}\frac{W_q}{s + \frac{\Delta_\phi}{2} - \tau - q}\right),
$$
(4.1)

with

$$
W_q := \frac{\Gamma^2(\tau + \ell + \Delta_\phi - h)}{\Gamma(2\Delta_\phi + \ell - h)}\frac{(1 + \tau - \Delta_\phi)_q^2}{\Gamma(q+1)\Gamma(2\tau + \ell - h + 1 + q)}.
$$
(4.2)

Now we will investigate a very specific limit. We will particularly look into the limit when $\tau \gg 1$, $\Delta_\phi \gg 1$. The flat space limit makes it necessary to consider $\Delta_\phi \gg 1$. Why we are considering $\tau \gg 1$ will become clear in a moment. We will also consider $\ell \gg 1$. The last assumption is for now a working assumption which will be justified in due course[5].

Now in the limit that $\Delta_\phi \gg 1$, $\tau \gg 1$ the residue function $W_q$ is peaked around $q = q_\star \sim O(\tau)$ Such an observation was first made in [15]. In fact, in this limit we can approximate the residue by a Gaussian function[6],

$$
W_q \approx \frac{1}{\sqrt{2\pi}(\ell + 2\Delta_\phi)\delta q}\, e^{-\frac{(q - q_\star)^2}{2\delta q^2}},
$$
(4.3)

with

$$
\begin{aligned}
q_\star &= \frac{(\tau - \Delta_\phi)^2}{\ell + 2\Delta_\phi},\\
\delta q^2 &= \frac{(\tau - \Delta_\phi)^2(\ell + \tau + \Delta_\phi)^2}{(\ell + 2\Delta_\phi)^3}.
\end{aligned}
$$
(4.4)

From this above expression, note that while $q_\star \sim O(\tau)$ in the limit of large $\tau$ one has $\delta q \sim O(\sqrt{\tau})$ in the same limit. This suggests that in the limit $\tau \to \infty$ we can in fact consider the above Gaussian as a Dirac Delta function to leading order. To see this explicitly, we introduce the "normalized variable",

$$
\bar{q} = \frac{q}{q_\star}.
$$
(4.5)

Now we define

$$
\epsilon := \left(\frac{\delta q}{q_\star}\right)^2.
$$
(4.6)

In these new variables $\bar{q}, \epsilon$ we have ,

$$
W_q \approx \frac{1}{q_\star(\ell + 2\Delta_\phi)}\frac{e^{-\frac{(\bar{q}-1)^2}{2\epsilon}}}{\sqrt{2\pi\epsilon}}.
$$
(4.7)

Further note that

$$
\frac{\delta q}{q_\star} \sim O(\tau^{-1/2}), \quad \tau \to \infty,
$$
(4.8)

which further implies the equivalence of the limits $\tau \to \infty$ and $\epsilon \to 0$. Thus in this limit,

$$
\lim_{\epsilon \to 0} W_q \approx \frac{1}{q_\star(\ell + 2\Delta_\phi)}\lim_{\epsilon \to 0}\frac{e^{-\frac{(\bar{q}-1)^2}{2\epsilon}}}{\sqrt{2\pi\epsilon}} = \frac{1}{q_\star(\ell + 2\Delta_\phi)}\,\delta(\bar{q} - 1) = \frac{1}{\ell + 2\Delta_\phi}\,\delta(q - q_\star).
$$
(4.9)

---

[5]In particular, the working assumption that, $\ell \gg 1$ has really *nothing* to do with flat space limit.

[6]Here, we would like to mention that, this approximated expression is obtained by implicitly considering $\Delta_\phi \sim \tau$ along with $\Delta_\phi \gg 1$, $\tau \gg 1$.

Now we can use this to write the $q-$sum as,

$$\sum_{q=0}^{\infty} \frac{W_q}{s + \frac{\Delta_\phi}{2} - \tau - q} \approx \int dq \frac{W_q}{s + \frac{\Delta_\phi}{2} - \tau - q} \approx \frac{1}{(\ell + 2\Delta_\phi)\left(s + \frac{\Delta_\phi}{2} - \tau - q_\star\right)}. \tag{4.10}$$

Since we are considering the limit $\tau \gg 1$ and $s \gg \tau \gg 1$ we can now use the Gegenbauer asymptotic of the Mack polynomials which is worked out in appendix A. Using this, we have

$$\mathcal{M}(s,t) \approx -\sum_{\tau,\ell} C_{\tau,\ell} \mathcal{N}_{\tau,\ell} \frac{\Gamma(2\Delta_\phi + \ell - h)}{2\Delta_\phi + \ell} \frac{\sin^2 \pi\left[\frac{\Delta_\phi}{2} - s\right]}{\sin^2 \pi\left[\Delta_\phi - \tau - \frac{\ell}{2}\right]}$$
$$\left(\frac{s}{8}\right)^\ell \frac{\Gamma(\ell+1)}{(h-1)_\ell} C_\ell^{(h-1)}(x) \left(\frac{1}{s + \frac{\Delta_\phi}{2} - \tau - q_\star}\right), \tag{4.11}$$

with

$$x = 1 + \frac{2t}{s - \Delta_\phi/2}. \tag{4.12}$$

Now recalling that,

$$\mathcal{A}_M(s,t) = \text{Im}_s \mathcal{M}(s,t) \tag{4.13}$$

we have

$$\mathcal{A}_M(s,t) \approx \pi \sum_{\tau,\ell} C_{\tau,\ell} \mathcal{N}_{\tau,\ell} \frac{\Gamma(2\Delta_\phi + \ell - h)}{2\Delta_\phi + \ell} \frac{\sin^2 \pi\left[\frac{\Delta_\phi}{2} - s\right]}{\sin^2 \pi\left[\Delta_\phi - \tau - \frac{\ell}{2}\right]}$$
$$\left(\frac{s}{8}\right)^\ell \frac{\Gamma(\ell+1)}{(h-1)_\ell} C_\ell^{(h-1)}(x) \, \delta\left(s + \frac{\Delta_\phi}{2} - \tau - q_\star\right). \tag{4.14}$$

Now ultimately we are interested in quantities which are integrals of $\mathfrak{a}_{n,\Delta_\phi}$ and $\bar{\mathcal{A}}_M(s)$. This integral over $s$ effectively truncates the $\tau-$ sum due to presence of the Dirac delta function. As a consequence of this we have the following expression, obtained in the forward limit $x \to 1$[7]

$$\bar{\mathcal{A}}_M(s) \approx \frac{2\pi}{2s - \Delta_\phi} \sum_{\substack{\ell \\ \ell \text{ even}}} \frac{(2h-2)_\ell}{(h-1)_\ell} \frac{\Gamma(2\Delta_\phi + \ell - h)}{2\Delta_\phi + \ell} \sum_{\tau=\Delta_\phi}^{\tau_\star} \left(\frac{1}{8}\right)^\ell C_{\tau,\ell} \, \mathcal{N}_{\tau,\ell} \left(\tau + q_\star - \frac{\Delta_\phi}{2}\right)^\ell, \tag{4.15}$$

where $\tau_\star$ satisfies,

$$\tau_\star + q_\star(\tau_\star) = s + \frac{\Delta_\phi}{2}. \tag{4.16}$$

Solving the equation and choosing the positive root for $\tau_\star$,

$$\tau_\star = \frac{1}{2}\left(\sqrt{(2\Delta_\phi + \ell)(\ell + 4s)} - \ell\right). \tag{4.17}$$

Now assuming $s \gg \ell$ we can approximate,

$$\tau_\star \approx \sqrt{(2\Delta_\phi + \ell)s}. \tag{4.18}$$

---

[7]In obtaining eq.(4.15), we have made explicit use of the fact that, operators with even spin ($\ell$), only, gets exchanged in the OPE channels of identical scalars. We have also used the fact that, $q_\star$ is an integer. Using these, one finds that the term $\sin^2 \pi[\Delta_\phi/2 - s]/\sin^2 \pi[\Delta_\phi - \tau - \ell/2]$ becomes unity on doing the $s$ integral in obtaining eq.(4.15).

Thus, we will consider[8]

$$\bar{\mathcal{A}}_M(s) \approx \frac{2\pi}{2s - \Delta_\phi} \sum_{\substack{\ell \\ \ell \text{ even}}} \frac{(2h-2)_\ell}{(h-1)_\ell} \frac{\Gamma(2\Delta_\phi + \ell - h)}{2\Delta_\phi + \ell} \sum_{\tau = \Delta_\phi}^{\sqrt{(2\Delta_\phi + \ell)s}} \left(\frac{1}{8}\right)^\ell C_{\tau,\ell} \, \mathcal{N}_{\tau,\ell} \left(\tau + q_\star - \frac{\Delta_\phi}{2}\right)^\ell .$$

(4.19)

Now observe that,

$$q_\star = \frac{(\tau - \Delta_\phi)^2}{\ell + 2\Delta_\phi} \leq \frac{(\tau - \Delta_\phi)^2}{2\Delta_\phi}.$$

(4.20)

Further using this we can write,

$$\left(\tau + q_\star - \frac{\Delta_\phi}{2}\right)^\ell \leq \left[\tau + \frac{(\tau - \Delta_\phi)^2}{2\Delta_\phi} - \frac{\Delta_\phi}{2}\right]^\ell = \left(\frac{\tau^2}{2\Delta_\phi}\right)^\ell$$

(4.21)

because we have $\ell \geq 0$. Then using this we can write

$$\bar{\mathcal{A}}_M(s) \leq \frac{2\pi}{2s - \Delta_\phi} \sum_{\substack{\ell \\ \ell \text{ even}}} \frac{(2h-2)_\ell}{(h-1)_\ell} \frac{\Gamma(2\Delta_\phi + \ell - h)}{2\Delta_\phi + \ell} \sum_{\tau = \Delta_\phi}^{\sqrt{(2\Delta_\phi + \ell)s}} \left(\frac{\tau^2}{16\Delta_\phi}\right)^\ell C_{\tau,\ell} \, \mathcal{N}_{\tau,\ell} .$$

(4.22)

### 4.1.1 Determining the $\ell$−cutoff

Now we move on to the determination of the cutoff for the $\ell$−sum in the expression for $\bar{\mathcal{A}}_M(s)$. To do so we will take help of the "polynomial boundedness" condition that is expressed through the finiteness of the integral quantity $\mathfrak{a}_{n,\rho}$ for some positive integer $n$. Now since $\mathcal{A}_M(s,t)$ is a positive distribution for unitary theories, we can write the following the chain of inequalities,

$$\mathfrak{a}_{n,\rho} = \int_{\frac{\Delta_\phi}{2}}^{\infty} \frac{d\bar{s}}{\bar{s}^{n+1}} \mathcal{A}_M\left(\bar{s}, t = \frac{\rho \Delta_\phi}{2}\right) > \int_{\frac{\Delta_\phi}{2}}^{s} \frac{d\bar{s}}{\bar{s}^{n+1}} \mathcal{A}_M\left(\bar{s}, t = \frac{\rho \Delta_\phi}{2}\right)$$

$$\geq s^{-(n+1)} \int_{\frac{\Delta_\phi}{2}}^{s} d\bar{s} \, \mathcal{A}_M\left(\bar{s}, t = \frac{\rho \Delta_\phi}{2}\right),$$

(4.23)

where the last inequality was possible because $n \geq 0$. Thus we have the following inequality,

$$\mathfrak{a}_{n,\rho} \geq \pi s^{-(n+1)} \sum_{\substack{\ell \\ \ell \text{ even}}} \frac{\Gamma(\ell+1)}{(h-1)_\ell} \frac{\Gamma(2\Delta_\phi + \ell - h)}{2\Delta_\phi + \ell} C_\ell^{(h-1)} \left(1 + \frac{\rho \Delta_\phi}{s - \Delta_\phi/2}\right)$$

$$\sum_{\tau = \Delta_\phi}^{\tau_\star} \left(\frac{1}{8}\right)^\ell C_{\tau,\ell} \mathcal{N}_{\tau,\tilde{\ell}} \left(\tau + q_\star - \frac{\Delta_\phi}{2}\right)^\ell$$

---

[8]In footnote 6 it was mentioned that, the analysis so far was carried out by implicitly considering $\Delta_\phi \sim \tau$ along with $\Delta_\phi \gg 1$, $\tau \gg 1$. However, observe that, the upper limit of the $\tau$−sum really does not conform to $\tau \sim \Delta_\phi$. In the upper limit one has, in fact, $\tau \gg \Delta_\phi$. But, this does not cause any issue because, the center of our subsequent analysis, eq.(4.21), is really independent of this.

$$\geq \pi s^{-(n+1)} \sum_{\substack{\ell=L+2 \\ \ell \text{ even}}} \frac{\Gamma(\ell+1)}{(h-1)_\ell} \frac{\Gamma(2\Delta_\phi+\ell-h)}{2\Delta_\phi+\ell} C_\ell^{(h-1)}\left(1+\frac{\rho\Delta_\phi}{s-\Delta_\phi/2}\right)$$

$$\sum_{\tau=\Delta_\phi}^{\tau_\star} \left(\frac{1}{8}\right)^\ell C_{\tau,\ell} \mathcal{N}_{\tau,\tilde{\ell}} \left(\tau+q_\star-\frac{\Delta_\phi}{2}\right)^\ell$$

$$\geq \pi s^{-(n+1)} \mathbf{C}_{L+2}^{(h-1)}\left(1+\frac{\rho\Delta_\phi}{s-\Delta_\phi/2}\right)$$

$$\sum_{\substack{\ell=L+2 \\ \ell \text{ even}}} \frac{(2h-2)_\ell}{(h-1)_\ell} \frac{\Gamma(2\Delta_\phi+\ell-h)}{2\Delta_\phi+\ell} \sum_{\tau=\Delta_\phi}^{\tau_\star} \left(\frac{1}{8}\right)^\ell C_{\tau,\ell} \mathcal{N}_{\tau,\tilde{\ell}} \left(\tau+q_\star-\frac{\Delta_\phi}{2}\right)^\ell,$$

where $\mathbf{C}_\ell^{(\alpha)}$ is the normalized Gegenbauer polynomial

$$\mathbf{C}_\ell^{(\alpha)}(x) = \frac{C_\ell^{(\alpha)}(x)}{C_\ell^{(\alpha)}(1)} = \frac{\Gamma(\ell+1)}{(2\alpha)_\ell} C_\ell^{(\alpha)}(x). \tag{4.24}$$

$L$ is some value of $\ell$ which is to be determined later and the last inequality is obtained using the fact that for $C_\ell^{(h-1)}(x)$ is an increasing function of $\ell$ for $x > 1$ and also accounting for the correct normalization of the Gegenbauer polynomial.

Now we can split the sum in eq.(4.15) in the following manner,

$$\bar{\mathcal{A}}_M(s) \approx \frac{2\pi}{2s-\Delta_\phi} \sum_{\substack{\ell \\ \ell \text{ even}}}^{L} \frac{(2h-2)_\ell}{(h-1)_\ell} \frac{\Gamma(2\Delta_\phi+\ell-h)}{2\Delta_\phi+\ell} \sum_{\tau=\Delta_\phi}^{\tau_\star} \left(\frac{1}{8}\right)^\ell C_{\tau,\ell} \mathcal{N}_{\tau,\ell} \left(\tau+q_\star-\frac{\Delta_\phi}{2}\right)^\ell + \mathcal{R}(s), \tag{4.25}$$

where

$$\mathcal{R}(s) = \frac{2\pi}{2s-\Delta_\phi} \sum_{\substack{\ell=L+2 \\ \ell \text{ even}}}^{\infty} \frac{(2h-2)_\ell}{(h-1)_\ell} \frac{\Gamma(2\Delta_\phi+\ell-h)}{2\Delta_\phi+\ell} \sum_{\tau=\Delta_\phi}^{\tau_\star} \left(\frac{1}{8}\right)^\ell C_{\tau,\ell} \mathcal{N}_{\tau,\ell} \left(\tau+q_\star-\frac{\Delta_\phi}{2}\right)^\ell. \tag{4.26}$$

Quite obviously, then, we can write

$$\mathcal{R}(s) \leq \frac{\mathfrak{a}_{n,\rho}\, s^{n+1}}{(2s-\Delta_\phi)\, \mathbf{C}_L^{(h-1)}\left(1+\frac{\rho\Delta_\phi}{s-\Delta_\phi/2}\right)}. \tag{4.27}$$

For $s \gg \Delta_\phi$ the above inequality effectively is,

$$\mathcal{R}(s) \leq \frac{\mathfrak{a}_{n,\rho} s^n}{\mathbf{C}_L^{(h-1)}\left(1+\frac{\rho\Delta_\phi}{s-\Delta_\phi/2}\right)}. \tag{4.28}$$

Next will make use of the following bounding relation satisfied by the Gegenbauer polynomials (see appendix B for a derivation),

$$\mathbf{C}_\ell^{(\alpha)}(z) \geq 2^{1-2\alpha} \frac{\Gamma(2\alpha)}{\Gamma^2(\alpha)} K(\varphi_0) \left(z+\sqrt{z^2-1}\cos\varphi_0\right)^\ell, \tag{4.29}$$

with

$$K(\varphi_0) = \int_0^{\varphi_0} (\sin\varphi)^{2\alpha-1} d\varphi$$

for any $\varphi_0$, $0 < \varphi_0 < \pi$, $x > 1$, $\alpha > 0$. Employing this we can constrain $\mathcal{R}(s)$ as,

$$\mathcal{R}(s) \le 2^{3-2h} \, \mathfrak{a}_{n,\rho} s^n \frac{\Gamma(2h-2)}{\Gamma^2(h-1)} \left( \tilde{x} + \sqrt{\tilde{x}^2 - 1} \cos \varphi_0 \right)^{-L-2}, \tag{4.30}$$

with

$$\tilde{x} := 1 + \frac{\rho \Delta_\phi}{s - \Delta_\phi/2}. \tag{4.31}$$

Now for $s \gg \Delta_\phi$ we have to leading order,

$$\left( \tilde{x} + \cos \varphi_0 \sqrt{\tilde{x}^2 - 1} \right) \sim 1 + \cos \varphi_0 \sqrt{\frac{2\rho \Delta_\phi}{s}}. \tag{4.32}$$

Thus we can write ,

$$\mathcal{R}(s) \le 2^{3-2h} \, \mathfrak{a}_{n,\rho} s^n \frac{\Gamma(2h-2)}{\Gamma^2(h-1)} \left( 1 + \cos \varphi_0 \sqrt{\frac{2\rho \Delta_\phi}{s}} \right)^{-L-2}. \tag{4.33}$$

Now the optimal value for $L$ can be obtained by demanding that the remainder term be of exponentially suppressed magnitude. However there is a subtlety in this requirement. The important thing to keep in mind is that we need to have the remainder exponentially suppressed compared to the truncated sum eq.(4.22). What this means is that we are keeping the possibility of certain overall growth behaviour (that of the truncated sum) for the remainder term but still sticking to the requirement that the growth be multiplied by a strong exponential suppression. Thus we are making the requirement a bit weaker than eq.(4.33). Assume a polynomial behaviour for the truncated sum $\sim s^a$ (here logarthmic terms may be present which we are ignoring because they are in general much weaker than a polynomial behaviour). Then the optimal $L$ is given by the rather weaker constraint ,

$$\mathcal{R}(s) \le 2^{3-2h} \, \mathfrak{a}_{n,\rho} \, s^{n-a} \frac{\Gamma(2h-2)}{\Gamma^2(h-1)} \left( 1 + \cos \varphi_0 \sqrt{\frac{2\rho \Delta_\phi}{s}} \right)^{-L-2}. \tag{4.34}$$

The optimal $L$ is thus given to leading order by,

$$\boxed{L = \frac{(n-a)}{\cos \varphi_0} \sqrt{\frac{s}{2\rho \Delta_\phi}} \ln s.} \tag{4.35}$$

We will truncate the $\ell$−sum in eq.(4.22) at $\ell = L$ as determined above to obtain the asymptotic bound,

$$\bar{\mathcal{A}}_M(s) \le \frac{2\pi}{2s - \Delta_\phi} \sum_{\substack{\ell \\ \ell \text{ even}}}^{L} \frac{(2h-2)_\ell}{(h-1)_\ell} \frac{\Gamma(2\Delta_\phi + \ell - h)}{2\Delta_\phi + \ell} \sum_{\tau=\Delta_\phi}^{\sqrt{(2\Delta_\phi + \ell)s}} \left( \frac{\tau^2}{16\Delta_\phi} \right)^\ell C_{\tau,\ell} \, \mathcal{N}_{\tau,\ell}. \tag{4.36}$$

But to achieve the main goal of bounding $\bar{\mathcal{A}}_M(s)$, we will need to have some information about the $\tau$−sum appearing as in the above expression. This is what we turn to next.

### 4.1.2 The final bounds

In order to obtain the final bounding expression, we need to have an estimate of the sum over $\tau$ of $C_{\tau,\ell} \mathcal{N}_{\tau,\ell}$. We are concerned with the large $s$ asymptotic of the sum,

$$\sum_{\tau=\Delta_\phi}^{\sqrt{(2\Delta_\phi + \ell)s}} \left( \frac{\tau^2}{16\Delta_\phi} \right)^\ell C_{\tau,\ell} \mathcal{N}_{\tau,\ell}. \tag{4.37}$$

It should be possible to do an analysis of this sum using the complex Tauberian theorem arguments used in [17]. However, we will content ourselves using a weaker result for now. To obtain the leading term in the asymptotic, we consider the generalized mean field theory (MFT) value for $C_{\tau,\ell}$ and consider the large $\tau$ limit of the same. The reason behind this is that the MFT operators are needed to reproduce the identity exchange in the crossed channel. This result is valid for spins greater than 2 and is a general result derived in [48]. The large $\Delta_\phi$ limit that we consider does not affect the conclusions. Thus our results should be valid in any CFT with the identity operator.

So we consider the large $\tau$−limit of the product,

$$\left(\frac{\tau^2}{16\Delta_\phi}\right)^\ell C_{\tau,\ell} \mathcal{N}_{\tau,\ell} \sim \frac{2^{2h+1}\tau^{4-2h}(2\Delta_\phi)^{-\ell}\,\Gamma(\ell+h)}{\pi^2\Gamma^2(\Delta_\phi)\Gamma(\ell+1)\Gamma^2(-h+\Delta_\phi+1)}\sin^2\pi[\Delta_\phi-\tau]. \tag{4.38}$$

Now at this point we have two separate cases at hand. As shown in appendix E the sum eq.(4.37) above has different asymptotes depending upon whether $h$ is greater, equal or less than 5/2. We have

$$\frac{\pi^2\Gamma^2(\Delta_\phi)\Gamma(\ell+1)\Gamma^2(-h+\Delta_\phi+1)}{2^{2h+1}(2\Delta_\phi)^{-\ell}\,\Gamma(\ell+h)}$$

$$\times \sum_{\tau=\Delta_\phi}^{\sqrt{(2\Delta_\phi+\ell)s}}\left(\frac{\tau^2}{16\Delta_\phi}\right)^\ell C_{\tau,\ell}\mathcal{N}_{\tau,\ell} \sim \begin{cases} \frac{1}{10-4h}\left[s(2\Delta_\phi+\ell)\right]^{\frac{5}{2}-h}, & h<\frac{5}{2}; \\[2mm] \frac{1}{4}\log s, & h=\frac{5}{2}; \\[2mm] \frac{\Delta_\phi^{5-2h}}{4h-10}, & h>\frac{5}{2}. \end{cases} \tag{4.39}$$

Now with the aid of this expression we turn to the final step of obtaining the Froissart bound for the Mellin amplitude.

**Case I.** $h<\frac{5}{2}$

First we start with the case $h<5/2$. Taking the large $\ell$ asymptotic of eq.(4.39) for $h<5/2$ we have ,

$$\sum_{\tau=\Delta_\phi}^{\sqrt{(2\Delta_\phi+\ell)s}}\left(\frac{\tau^2}{16\Delta_\phi}\right)^\ell C_{\tau,\ell}\mathcal{N}_{\tau,\ell} \sim s^{\frac{5}{2}-h}\,\ell^{h-1}\frac{2^{2h+1}(2\Delta_\phi+\ell)^{\frac{5}{2}-h}(2\Delta_\phi)^{-\ell}}{\pi^2(10-4h)\,\Gamma^2(\Delta_\phi)\Gamma^2(-h+\Delta_\phi+1)}. \tag{4.40}$$

Next putting this into eq.(4.22),

$$\bar{A}_M(s)\leq \frac{2\pi}{2s-\Delta_\phi}s^{\frac{5}{2}-h}\sum_{\substack{\ell=0 \\ \ell\ \text{even}}}^{L}\frac{2^{2h+1}(2\Delta_\phi+\ell)^{\frac{3}{2}-h}\,\Gamma(2\Delta_\phi-h)(2\Delta_\phi-h)_\ell}{\pi^2(10-4h)\Gamma^2(\Delta_\phi)\Gamma^2(-h+\Delta_\phi+1)(2\Delta_\phi)^\ell}\frac{(2h-2)_\ell}{(h-1)_\ell}\ell^{h-1}, \tag{4.41}$$

where we have used $\Gamma(2\Delta_\phi-h+\ell)=(2\Delta_\phi-h)_\ell\Gamma(2\Delta_\phi-h)$. Next, we will consider the large $\ell$ asymptotic[9]

$$\frac{(2h-2)_\ell}{(h-1)_\ell}\sim \ell^{h-1}\frac{\Gamma(h-1)}{\Gamma(2h-2)}. \tag{4.42}$$

Now we consider that $\Delta_\phi\gg h$, $\Delta_\phi\gg 1$. At this point, to make progress (for $d=3$ we do not have to make this choice), we approximate $2\Delta_\phi+\ell\sim 2\Delta_\phi$ by assuming[10] $\Delta_\phi\gg L$. Then one

---

[9]The dominant contribution to the $\ell$-sum comes from the upper limit $\ell=L$ and since, $L$ is large we have used large $\ell$ approximation of for the $\ell$-summand. We observe that, this works because the $\ell$-summand behaves as power law with positive exponent in the large $\ell$ limit.

[10]This follows from the discussion in section 3.3. The (very interesting) case where $s\gg(2\Delta_\phi)^3$ and which will make a difference for $d\neq 3$ is beyond the scope of this work.

obtains,

$$\bar{\mathcal{A}}_M(s) \leq s^{\frac{3}{2}-h} \frac{2^{2h}\Gamma(2\Delta_\phi - h)}{\pi(5-2h)\Gamma^2(\Delta_\phi)\Gamma^2(-h+\Delta_\phi+1)} \frac{\Gamma(h-1)}{\Gamma(2h-2)} \sum_{\substack{\ell=0 \\ \ell \text{ even}}}^{L} \ell^{2h-2} (2\Delta_\phi)^{\frac{3}{2}-h}, \quad (4.43)$$

where we have used $s \gg \Delta_\phi/2$.

We can now use eq.(F.12) to obtain

$$\bar{\mathcal{A}}_M(s) \leq \frac{2^{2h}(2\Delta_\phi)^{\frac{3}{2}-h}\Gamma(2\Delta_\phi-h)}{\pi\Gamma^2(\Delta_\phi)\Gamma^2(-h+\Delta_\phi+1)} \frac{\Gamma(h-1)}{(5-2h)\Gamma(2h-2)} s^{\frac{3}{2}-h} \frac{L^{2h-1}}{4h-2}. \quad (4.44)$$

Now using eq.(4.35) we have,

$$\boxed{\bar{\mathcal{A}}_M(s) \leq \mathcal{B}_1 s \ln^{2h-1} s, \quad (4.45)}$$

with

$$\mathcal{B}_1 = 2^{2h-1}(2\Delta_\phi)^{2-2h} \frac{8\pi^{h-1}\mathcal{N}\,\Gamma(h-1)}{(5-2h)(2h-1)\Gamma(2h-2)} \left(\frac{n-1}{\sqrt{\rho}\cos\varphi_0}\right)^{2h-1}, \quad (4.46)$$

where $\mathcal{N}$ is same as in eq.(3.18) and we have put $a = 1$ by observing that the leading power law dependency of bound is $\sim s$.

$\underline{d=2}$ : At this point we would like to comment upon the case of $d = 2$, or equivalently $h = 1$. Note that in this case the Gegenbauer polynomial $C_\ell^{(h-1)}$ is undefined. But this case can still be tackled following the analysis of [10]. In fact on following the method one obtains the bound in this case coincident with eq.(4.45) if we put $h = 1$ and $\cos\varphi_0 = 1$ formally into the same. Note that, while formally putting $h = 1$ into eq.(4.45) one has to consider doing so in the limiting sense if required.

**Case II:** $h = \frac{5}{2}$

Next we turn to the case $h = \frac{5}{2}$. Considering the large $\ell$ limit as before one readily obtains from eq.(4.39)

$$\sum_{\tau=\Delta_\phi}^{\sqrt{(2\Delta_\phi+\ell)s}} \left(\frac{\tau^2}{16\Delta_\phi}\right)^\ell C_{\tau,\ell}\mathcal{N}_{\tau,\ell} \sim \frac{16\,\log(s)\,\ell^{\frac{3}{2}}}{\pi^2\Gamma^2\left(\Delta_\phi-\frac{3}{2}\right)\Gamma^2\left(\Delta_\phi\right)} (2\Delta_\phi)^{-\ell}. \quad (4.47)$$

Thus we have,

$$\bar{\mathcal{A}}_M(s) \leq \frac{\log s}{s} \frac{16\Gamma(2\Delta_\phi - 5/2)}{2\pi\Delta_\phi\Gamma^2(\Delta_\phi - 3/2)\Gamma^2(\Delta_\phi)} \sum_{\ell=0}^{L}\ell^3 \sim \frac{\log s}{s} \frac{16\Gamma(2\Delta_\phi-5/2)}{2\pi\Delta_\phi\Gamma^2(\Delta_\phi-3/2)\Gamma^2(\Delta_\phi)} \frac{L^4}{8}, \quad (4.48)$$

where the last equality follows by the large $L$ asymptotic of the $\ell-$sum. Next using eq.(4.35),

$$\boxed{\bar{\mathcal{A}}_M(s) \leq \mathcal{B}_2 s \log^5 s, \quad (4.49)}$$

with

$$\mathcal{B}_2 := 8\pi^{\frac{3}{2}}\mathcal{N}(2\Delta_\phi)^{-3} \left(\frac{(n-1)}{\sqrt{\rho}\cos\varphi_0}\right)^4, \quad (4.50)$$

where $a = 1$ has been put in the last stage by the same logic as in the previous case.

**Case III:** $h > \frac{5}{2}$

Now we turn to the case when $h > 5/2$. This is rather curious case. As shown in appendix E, for this case the lower limit of the sum eq.(4.37) dominates rather than the upper limit. As a consequence, we now have,

$$\sum_{\tau=\Delta_\phi}^{\sqrt{(2\Delta_\phi+\ell)s}} \left(\frac{\tau^2}{16\Delta_\phi}\right)^\ell C_{\tau,\ell}\mathcal{N}_{\tau,\ell} \sim \frac{2^{4(h-1)}(2\Delta_\phi)^{5-2h-\ell}\Gamma(\ell+h)}{\pi^2(4h-10)\Gamma^2(\Delta_\phi)\Gamma(\ell+1)\Gamma^2(-h+\Delta_\phi+1)}, \quad s \to \infty. \tag{4.51}$$

Further considering large $\ell$ limit ,

$$\sum_{\tau=\Delta_\phi}^{\sqrt{(2\Delta_\phi+\ell)s}} \left(\frac{\tau^2}{16\Delta_\phi}\right)^\ell C_{\tau,\ell}\mathcal{N}_{\tau,\ell} \sim \ell^{h-1}\frac{2^{4h-4}(2\Delta_\phi)^{5-2h-\ell}}{\pi^2(4h-10)\Gamma^2(\Delta_\phi)\Gamma^2(-h+\Delta_\phi+1)}. \tag{4.52}$$

Further putting this into eq.(4.19) and following through the same steps as before we get the asymptotic bound,

$$\bar{\mathcal{A}}_M(s) \leq \frac{1}{s} \sum_{\substack{\ell=0 \\ \ell \text{ even}}}^{L} \frac{2^{4h-4}(2\Delta_\phi)^{4-2h}\Gamma(2\Delta_\phi-h)\Gamma(h-1)}{\pi(4h-10)\Gamma(2h-2)\Gamma^2(\Delta_\phi)\Gamma^2(-h+\Delta_\phi+1)}\ell^{2h-2}. \tag{4.53}$$

Now using eq.(F.5),

$$\bar{\mathcal{A}}_M(s) \leq \mathcal{B}_3\, s^{h-\frac{3}{2}}\ln^{2h-1}s\,, \tag{4.54}$$

with

$$\mathcal{B}_3 := 2^{4h-6}(2\Delta_\phi)^{\frac{9}{2}-3h}\frac{8\pi^{h-1}\mathcal{N}\,\Gamma(h-1)}{(2h-5)(2h-1)\Gamma(2h-2)}\left(\frac{2(n-h)+3}{\sqrt{\rho}\cos\varphi_0}\right)^{2h-1}. \tag{4.55}$$

### 4.1.3 On the number of subtractions of Mellin Amplitude dispersion relation

The polynomial boundedness assumption for the Mellin amplitude is closely tied to the question of writing a dispersion relation for the Mellin amplitude. The key point in this regard is how many subtractions are sufficient to write such a dispersion relation. The assumption of finiteness of $\mathfrak{a}_n$ naively suggests the possibility of writing a dispersion relation for Mellin amplitude with $n-$subtractions. Then the question is what can be the value of $n$. In the above analysis we have kept $n$ arbitrary. $n$ will be determined by the leading power law behaviour of the bound. What we mean by this is that the we have already seen that the generic structure of the Froissart$_{AdS}$ bound for $\bar{\mathcal{A}}_M(s)$ is of the form $\bar{\mathcal{A}}_M(s) \leq Cs^a \ln^b s$. Now it turns out that *the value of $n$ is controlled by $a$*. This control happens in two ways.

First observe the expression for the optimal value of the $\ell-$cutoff in eq.(4.35). There sits a factor of $(n-a)$. Now, in our analysis we have extensively used the assumption $L \gg 1$. Then for the consistency of this assumption we require necessarily $n > a$.

While this simple consideration puts a lower bound on the magnitude of $n$, it is also possible to obtain an upper bound on the same. The way to have so is by using a theorem from complex analysis called Phragmen-Lindeloff theorem (see, for example, [34]). The general logic goes as follows: assuming the polynomial boundedness condition as in section 3.5 and using the Froissart$_{AdS}$ bound it is possible to show by the use of Phragmen-Lindeloff theorem that $n \leq \lfloor a \rfloor + 1$. This thus puts an upper bound on $n$. Now we analyse the individual cases of different $h$ values.

I. $h < 5/2$: For $h < 5/2$ we have from eq.(4.45) $a = 1$. Then following logic chalked out above we have clearly $n = 2$.

II. $h = 5/2$: For this case as well we have $a = 1$ from eq.(4.49). Thus again we will have $n = 2$.

III. $h > 5/2$: This case is rather interesting. From eq.(4.54) we have $a = (2h-3)/2$ thus leading immediately to,

$$\frac{2h-3}{2} < n \leq \lfloor \frac{2h-3}{2} \rfloor + 1. \tag{4.56}$$

What this implies is that, while for $h = 3$ ( equivalently $d = 6$) one has to have $n = 2$, *one must have $n \geq 3$ for $h > 3$ i.e., $d > 6$. The number goes to infinity as $d$ goes to infinity.* In section 5, we will provide an alternative derivation of these results without invoking the Phragmen-Lindeloff theorem.

## 4.2 Connection to flat space Froissart bound

### 4.2.1 $d = 3$

Now that we have bounds on Mellin amplitude we would like to consider the flat space limit of the above bound. It is quite straightforward that $\mathcal{A}_M(s,t)$ is related to the absorptive part of the flat spacetime scattering amplitude $\mathcal{T}(S,T)$ as in eq.(3.17) and similar relations follow for all averaged quantities.

Now for the flat space limit we will focus our attention upon $h = 3/2$ i.e., $3d$ CFT which in the flat space limit connects to $(3+1)d$ flat spacetime quantum field theory where the original Froissart bound was proved. We start with the bounding relation eq.(4.45). Considering $n = 2$ and $\cos\varphi_0 = 1$ in a limiting sense for strongest bound in eq.(4.46) the Froissart$_{Ads}$ bound, eq.(4.45), becomes for $h = 3/2$,

$$\bar{\mathcal{A}}_M(s) \leq 4\pi \mathcal{N} \frac{s}{\rho \Delta_\phi} \ln^2 s. \tag{4.57}$$

Next taking the flat space limit and making use of eq.(3.20), one obtains[11],

$$\bar{\mathcal{A}}(S) \leq \frac{\pi}{2\rho m^2} S \ln^2 \frac{S}{S_0}. \tag{4.58}$$

The known bound on $\bar{\mathcal{A}}$ from literature is (recall we are averaging so there is an extra $1/2$),

$$\bar{\mathcal{A}}(S) \leq \frac{\pi}{2\mu^2} S \ln^2 \frac{S}{S_0}. \tag{4.59}$$

Thus what we find is an exact match provided we identify $\rho = \mu^2/m^2$, identifying $\mu$ as the lightest exchange in the t-channel. However here comes a crucial difference. One can check that using the MFT asymptotics, the sum over conformal partial waves converges[12] even for $\rho = 1$. This means we can set $\mu = m$, which is the mass of the external particle. If we do this, then in fact we will have better agreement with the existing numerical fits of the proton-proton data. This needs to be checked carefully of course, which we will leave for future work.

---

[11]$S_0$ here has been put in on dimensional grounds.

[12]In the large $\ell$ limit, one can show that the summand in the $\ell$-sum goes like $\ell^{-2\sqrt{s}\ell} s^\ell / \ell!$ for any $\rho$. We need $\rho \leq 1$ since the measure factor $\mu(s,t)$ in the Mellin representation has $\Gamma^2(\Delta_\phi/2 - t)$ so there is a double pole at $t = \Delta_\phi/2$.

### 4.2.2 $d \neq 3$

Now we consider the case of $d > 3$. Here we would like to make a comparison of the flat space limit of the Mellin amplitude bounding relation with standard result of bounds on flat space scattering amplitude in general spacetime dimensions [9] given in eq.(2.9). Upon comparison one finds that ratio of the the frontal coefficient that is obtained on taking the flat space limit of eq.(4.45) to the frontal coefficient that appears in eq.(2.9) is,

$$\frac{2}{(5-2h)}. \tag{4.60}$$

This coefficient is unity for $d = 3$. Further for $d = 4$ we find a weaker flat space bound by taking flat space limit of Mellin amplitude. For $d = 2$ it is stronger.

### 4.2.3 On flat space limit of eq.(4.49) and eq.(4.54)

We can consider taking flat space limit of the Froissart$_{AdS}$ bound for $h \geq 5/2$, eq.(4.49) and eq.(4.54), using the dictionary eq.(3.17).

   I. $\underline{h = 5/2}$ : Upon applying the flat space limit translation on eq.(4.49) one obtains,

$$\bar{\mathcal{A}}(S) \leq \frac{\pi^{\frac{3}{2}}}{8m^2} \left( \frac{n-1}{\sqrt{\rho} \cos \varphi_0} \right)^4 \left( \frac{S}{m^2} \right) \ln^5 \frac{S}{S_0}. \tag{4.61}$$

   Recall that this supposed to be corresponding to flat space scattering in 6 spacetime dimensions. Now if we compare the above with standard Froissart-Martin bound in 6 spacetime dimensions (c.f. eqn (24) of [9]) for the dependency upon the Mandelstam variable $S$ then we realize that the bound eq.(4.61) above is a weaker one due to the presence of one extra power of $\ln S$.

  II. $\underline{h > 5/2}$ : Taking the flat space limit of eq.(4.54) yields ,

$$\bar{\mathcal{A}}(S) \leq \frac{2^{h+\frac{3}{2}} \pi^{h-1} \Gamma(h-1)}{(2h-5)(2h-1)\Gamma(2h-2)} \left( \frac{2(n-h)+3}{\sqrt{\rho} \cos \varphi_0} \right)^{2h-1} \left( \frac{S}{m^2} \right)^{h-\frac{3}{2}} \ln^{2h-1} \frac{S}{S_0}. \tag{4.62}$$

   Now if one to compare the $S$ dependency of this bound with that of the standard Froissart-Martin bound one readily observes that the bound eq.(4.62) becomes weaker with increasing $h$.

### 4.2.4 Is the difference in form for $d > 4$ expected?

We can give a heuristic reason to justify, that a crossover at some value of $d$ is expected, in the behaviour of the Froissart bound. Froissart in his original paper and Feynman independently [35] had a heuristic argument for the $\ln^2$ behaviour. The argument goes as follows. Imagine that the interaction is well approximated by a Yukawa type potential

$$V \sim g \frac{e^{-\mu r}}{r}.$$

Now the maximum interaction happens when $ge^{-\mu r_*} \sim 1$, giving $r_* \sim \frac{\ln g}{\mu}$. Now assuming that the coupling $g$ depends on the energy $E$ polynomially, i.e., $g \propto E^N$ and also assuming that $\mu$ does not depend on $E$, we will find $r_* \sim \frac{\ln E}{\mu}$. Thus, the scattering cross-section in $d + 1$ dimensions is

$$\sigma \sim r_*^{d-1} \propto \ln^{d-1} E.$$

Now let us assume that this $\ln^{d-1}$ behaviour is to be expected (which is what flat space calculations give). In our calculation, since $L \sim \sqrt{s} \ln s$, this can happen from a factor of $L^{d-1}$. Now in Mellin space considerations, the extra powers of $s$ are given by the twist sum. If we assume that the asymptotic growth of twist is of the form $\tau^a$ (so that no extra powers of $\ln s$ can come from here), then from the upper limit of the $\tau$ integral we will get $s^{a/2+1/2}/(a+1)$. So the overall power of $s$ (taking into account the $1/s$ in the definition) in $\bar{A}_M$ is then

$$\frac{s^{\frac{a+d-2}{2}}}{a+1},$$

so that to match with the existing flat space answers in the literature we must have $a = 4-d$. This makes the denominator $5-d$ so that for $d \geq 5$ there is a change in behaviour than what is expected since here the dominant contribution comes from the lower limit of the twist integral, which is independent of $s$. This is essentially what we find.

### 4.3  Obtaining the Froissart$_{AdS}$ bound: "Non-forward limit"

In the previous section the we tackled the problem of obtaining an asymptotic upper bound to $\bar{A}_M(s,t)$ for $s$ large and $t=0$ i.e., the forward limit. Now we turn to the same problem for $t \neq 0$. In fact, the main task i.e., that of obtaining an $\ell-$ cutoff, is already done. The new piece of information that we need now is an upper bound for the Gegenbauer polynomial $C_\ell^{(\lambda)}(x)$ for $x \in [-1, 1]$. At this point it is worth of mentioning that we are considering "physical" values of $t$ i.e., $t < 0$.

Starting with eq.(4.14) we have,

$$\bar{A}_M(s,t) \equiv \bar{A}_M(s,x) \approx \frac{2\pi}{2s - \Delta_\phi} \sum_{\substack{\ell \\ \ell \text{ even}}} \frac{\Gamma(\ell+1)}{(h-1)_\ell} \frac{\Gamma(2\Delta_\phi + \ell - h)}{2\Delta_\phi + \ell} C_\ell^{(h-1)}(x)$$
$$\times \sum_{\tau = \Delta_\phi}^{\tau_\star} \left(\frac{1}{8}\right)^\ell C_{\tau,\ell} \, \mathcal{N}_{\tau,\ell} \left(\tau + q_\star - \frac{\Delta_\phi}{2}\right)^\ell. \qquad (4.63)$$

Below we will write $\bar{A}_M(s,t)$ and $\bar{A}_M(s,x)$ interchangebly. However we will later explain that there is a subtle difference between holding $t$ fixed and holding $x$ fixed while considering $s \to \infty$.

Start with the following inequlity for Jacobi polynomial [36],

$$P_\ell^{(\alpha,\beta)}(\cos\theta) < \frac{K}{\theta^{\alpha+1/2} \, \ell^{1/2}}, \quad \alpha \geq -\frac{1}{2}, \qquad (4.64)$$

where $K$ is a constant. Now using the definition of the Gegenbauer polynomial in terms of Jacobi polynomials,

$$C_\ell^{(\lambda)}(x) = \frac{(2\lambda)_\ell}{(\lambda + 1/2)_\ell} P_\ell^{(\lambda-1/2, \lambda-1/2)}(x), \qquad (4.65)$$

we obtain by eq.(4.64) above

$$C_\ell^{(\lambda)}(\cos\theta) < \frac{(2\lambda)_\ell}{(\lambda + 1/2)_\ell} \frac{K}{\theta^\lambda \, \ell^{1/2}}. \qquad (4.66)$$

Further considering the large $\ell$ limit,

$$C_\ell^{(\lambda)}(\cos\theta) < \widehat{K} \ell^{\lambda-1} \theta^{-\lambda}, \qquad (4.67)$$

with

$$\widehat{K} = \frac{\Gamma\left(\lambda + \frac{1}{2}\right)}{\Gamma(2\lambda)} K. \tag{4.68}$$

Now we can use this inequality eq.(4.67) into eq.(4.63) to obtain [putting $x = \cos\theta$][13],

$$\bar{\mathcal{A}}_M(s, \cos\theta) \le \widehat{K} \frac{2\pi}{2s - \Delta_\phi} \theta^{1-h} \sum_{\substack{\ell \\ \ell \text{ even}}} \frac{\Gamma(\ell+1)}{(h-1)_\ell} \ell^{h-2} \frac{(2\Delta_\phi - h)_\ell}{2\Delta_\phi + \ell} \Gamma(2\Delta_\phi - h)$$

$$\sum_{\tau=\Delta_\phi}^{\tau_\star} \left(\frac{1}{8}\right)^\ell C_{\tau,\ell} \mathcal{N}_{\tau,\ell} \left(\tau + q_\star - \frac{\Delta_\phi}{2}\right)^\ell$$

$$\approx \widehat{K} \frac{2\pi}{2s - \Delta_\phi} \theta^{1-h} \sum_{\substack{\ell \\ \ell \text{ even}}} \frac{\Gamma(\ell+1)}{(h-1)_\ell} \ell^{h-2} \frac{(2\Delta_\phi)^\ell}{2\Delta_\phi + \ell} \Gamma(2\Delta_\phi - h)$$

$$\sum_{\tau=\Delta_\phi}^{\tau_\star} \left(\frac{1}{8}\right)^\ell C_{\tau,\ell} \mathcal{N}_{\tau,\ell} \left(\tau + q_\star - \frac{\Delta_\phi}{2}\right)^\ell,$$

where we have considered the large $\Delta_\phi$ limit. Next mimicking the same steps as in forward limit we can use eq.(4.39) for the $\tau$ sum in the above. Thus again as before we have three cases depending upon the value of $h$. Further the optimal value of $L$ where the $\ell$-sum will be truncated is same as before i.e that given by eq.(4.35).

1) **Case I:** $h < \frac{5}{2}$

$$\bar{\mathcal{A}}_M(s, \cos\theta) \le K_1 \, s^{\frac{3}{2}-h} \, \theta^{1-h} \sum_{\substack{\ell \\ \ell \text{ even}}}^L \ell^{h-1} (2\Delta_\phi + \ell)^{\frac{3-2h}{2}}, \tag{4.69}$$

with

$$K_1 = 8\pi^h \mathcal{N} \frac{2^{2h} \, \Gamma(h-1)}{\pi(5-2h)\Gamma(2h-2)} \widehat{K}. \tag{4.70}$$

Now using the result eq.(F.13) we obtain,

$$\bar{\mathcal{A}}_M(s, \cos\theta) \le K_1 \, (2\Delta_\phi)^{\frac{3-2h}{2}} s^{\frac{3}{2}-h} \theta^{1-h} \frac{L^h}{2h}, \quad 2\Delta_\phi \gg L \gg 1. \tag{4.71}$$

Now using the optimal value of L eq.(4.35),

$$\bar{\mathcal{A}}_M(s, \cos\theta) \le \mathcal{C} \, s^{\frac{3-h}{2}} \, \ln^h s \, \theta^{1-h}, \tag{4.72}$$

with

$$\mathcal{C} = \frac{K_1}{4h} (2\Delta_\phi)^{\frac{3}{2}(1-h)} \left(\frac{n - 3(1-h)/2}{\sqrt{\rho} \cos\varphi_0}\right)^h, \tag{4.73}$$

where we have put the apt values of $a$ following the same logic as in the forward case. Now for fixed $t$ with $s \gg \Delta_\phi \gg 1$ we can rewrite the bounding expression above in terms of $t$ by using

$$\theta \approx 2\sqrt{\frac{|t|}{s}}. \tag{4.74}$$

---

[13]We have pulled out the $\theta^{1-h}$ factor outside the $\tau$ sum since for $d \ge 2$, it behaves like $s^a$ with $a > 0$ so we can replace the $s'$ dependence by $s$ at the level of the $s'$ integral.

Thus we have the bound,

$$\bar{\mathcal{A}}_M(s,t) \leq \mathcal{C}_1 \, s \ln^h s \, |t|^{\frac{1-h}{2}} , \tag{4.75}$$

where $\mathcal{C}_1 = 2^{1-h}\mathcal{C}$. Note that $|t|$ takes care of the fact we are considering $t < 0$.

Using the flat space limit dictionary eq.(3.17) obtain the following bound,

$$\mathcal{A}(S,T) \leq \mathcal{K}_1 \, m^{3-2h} \left(\frac{S}{m^2}\right) \ln^h \frac{S}{S_0} \left(\frac{|T|}{m^2}\right)^{\frac{1-h}{2}} , \tag{4.76}$$

where $\mathcal{K}_1$ is constant.

2) **Case II:** $h = \frac{5}{2}$

$$\bar{\mathcal{A}}_M(s,\cos\theta) \leq K_2 \, \frac{\ln s}{s} \, \theta^{-\frac{3}{2}} \sum_{\substack{\ell \\ \ell \text{ even}}}^{L} \ell^{3/2} , \tag{4.77}$$

with

$$K_2 = 32\pi^2 \frac{\mathcal{N}}{\Delta_\phi} \widehat{K} . \tag{4.78}$$

Now doing the $\ell$−sum,

$$\sum_{\substack{\ell \\ \ell \text{ even}}}^{L} \ell^{3/2} = 2\sqrt{2} H_{\frac{L}{2}}^{(-\frac{3}{2})} \sim \frac{L^{5/2}}{5}, \quad L \to \infty. \tag{4.79}$$

Putting this into eq.(4.77) we obtain,

$$\bar{\mathcal{A}}_M(s,\cos\theta) \leq \frac{K_2}{5} \left(\frac{n-1/4}{\sqrt{\rho}\cos\varphi_0}\right)^{5/2} (2\Delta_\phi)^{-\frac{5}{4}} s^{\frac{1}{4}} \ln^{\frac{7}{2}} s \, \theta^{-\frac{3}{2}} . \tag{4.80}$$

Again we can express $\theta$ in terms of $t$ to obtain,

$$\bar{\mathcal{A}}_M(s,t) \leq \mathcal{C}_2 \, s \ln^{\frac{7}{2}} s \, |t|^{-\frac{3}{4}} , \tag{4.81}$$

with

$$\mathcal{C}_2 = \frac{\pi^2 \mathcal{N}}{5} \left(\frac{2}{\Delta_\phi}\right)^{\frac{9}{4}} \left(\frac{n-1/4}{\sqrt{\rho}\cos\varphi_0}\right)^{5/2} \widehat{K} . \tag{4.82}$$

If we now consider taking the flat space limit of the above bound using eq.(3.17) then we get,

$$\bar{\mathcal{A}}(S,T) \leq \frac{\mathcal{K}_2}{m^2} \left(\frac{S}{m^2}\right) \ln^{\frac{7}{2}} S \left(\frac{|T|}{m^2}\right)^{-\frac{3}{4}} , \tag{4.83}$$

where

$$\mathcal{K}_2 = \frac{2^{\frac{9}{4}}\pi^2}{5} \left(\frac{n-1/4}{\sqrt{\rho}\cos\varphi_0}\right)^{5/2} \widehat{K} . \tag{4.84}$$

3) **Case III:** $h > \frac{5}{2}$

Finally we come to to case of $h > 5/2$. Going thorugh the same steps as before we reach the following bounding relation,

$$\bar{\mathcal{A}}_M(s,t) \le K_3 \, s^{h-\frac{3}{2}} \ln^h s \, |t|^{\frac{1-h}{2}} , \tag{4.85}$$

where $K_3$ is a constant.

# 5 Dispersion relations

In this section, we will follow [35] and write down dispersion relations for the Mellin amplitudes. The bounds derived in the previous section for the non-forward limit will prove useful here. We begin by writing an $N$-subtracted dispersion relation

$$\mathcal{M}(s,t) = \sum_{m=0}^{N-1} C_m(t) s^m + \frac{s^N}{\pi} \int_{\frac{\Delta_\phi}{2}}^{\infty} ds' \frac{\mathcal{A}_M^{(s)}(s',t)}{s'^N(s'-s)} + \frac{u^N}{\pi} \int_{\frac{\Delta_\phi}{2}}^{\infty} du' \frac{\mathcal{A}_M^{(u)}(u',t)}{u'^N(u'-u)} , \tag{5.1}$$

where $s+t+u = \Delta_\phi/2$ and $C_n(t)$'s are analytic in $t$ for $t < \Delta_\phi/2$. The number of subtractions is related to the number of $C_m(t)$'s that one will need to take as input. For the identical scalar case that we have been considering so far, $\mathcal{A}_M^{(s)}(u,t) = \mathcal{A}_M^{(u)}(u,t)$, but we will keep the discussion more general. The bounds in the previous section, although derived for $t < 0$ will continue to hold for $t > 0$ for sufficiently small[14] $t$. The bounds are of the form $\bar{\mathcal{A}}(s,t) \le K s^a \ln^b s |t|^{\frac{1-h}{2}}$. This suggests that there exists a $t = t_0$ with $0 < t_0 \ll \Delta_\phi/2$ such that $\mathcal{A}(s,t) \le c s^{n-1+\epsilon}$ can be used inside an integral with $\epsilon < 1$. For instance, for $h \le 5/2$ we have $n = 2$ while for $h > 5/2$ we have $n = \lfloor \frac{2h-3}{2} \rfloor + 1$. This means that for $0 \le t \le t_0$ we can write the dispersion relation

$$\mathcal{M}(s,t) = \sum_{m=0}^{n-1} C_m(t) s^m + \frac{s^n}{\pi} \int_{\frac{\Delta_\phi}{2}}^{\infty} ds' \frac{\mathcal{A}_M^{(s)}(s',t)}{s'^n(s'-s)} + \frac{u^n}{\pi} \int_{\frac{\Delta_\phi}{2}}^{\infty} du' \frac{\mathcal{A}_M^{(u)}(u',t)}{u'^n(u'-u)} . \tag{5.2}$$

Comparing eq.(5.1) and eq.(5.2)) (assuming $N > n$) we get an equation

$$\sum_{m=0}^{N-1} C_m(t) s^m = \sum_{m=0}^{n-1} C_m(t) s^m + \sum_{m=n}^{N-1} \left( \frac{s^m}{\pi} \int_{\frac{\Delta_\phi}{2}}^{\infty} ds' \frac{\mathcal{A}_M^{(s)}(s',t)}{s'^{m+1}} + \frac{u^m}{\pi} \int_{\frac{\Delta_\phi}{2}}^{\infty} du' \frac{\mathcal{A}_M^{(u)}(u',t)}{u'^{m+1}} \right) . \tag{5.3}$$

Comparing the highest power of $s$ for large $s$ (assuming $N$ is even), we have

$$C_{N-1}(t) = \frac{1}{\pi} \int_{\frac{\Delta_\phi}{2}}^{\infty} ds' \frac{\mathcal{A}_M^{(s)}(s',t)}{s'^N} + \frac{1}{\pi} \int_{\frac{\Delta_\phi}{2}}^{\infty} du' \frac{\mathcal{A}_M^{(u)}(u',t)}{u'^N} . \tag{5.4}$$

We can Taylor expand the integrand around $t = 0$ writing $\mathcal{A}_M^{(s)}(s',t) = \sum_{k=0}^{\infty} A_n^{(s)}(s') t^n$ and $\mathcal{A}_M^{(u)}(u',t) = \sum_{k=0}^{\infty} A_n^{(u)}(u') t^n$ where it can be shown that the coefficients are all positive which follows from $\frac{d^n C_\ell^{(\lambda)}(x)}{dx^n} \ge 0$. Using this and the fact that $C_{N-1}(t)$ was analytic for $t < \Delta_\phi/2$, it follows that each integral on the rhs of eq.(5.4) is finite for $t < \Delta_\phi/2$. This in turn implies that

$$\frac{|\mathcal{M}(s,t)|}{s^N} \to 0, \tag{5.5}$$

---

[14]This can be explicitly checked using the expressions in [35] where an alternative derivation can be found.

as $s \to \infty$ for $t < \Delta_\phi / 2$. As a result we can consider one less subtraction in eq.(5.1) than what we started off with. For $N$ odd, the situation is similar with the number of subtractions going down by two. This can be repeated until we reach the conclusion that

$$\int_{\frac{\Delta_\phi}{2}}^{\infty} ds' \frac{\mathcal{A}_M^{(s)}(s', t)}{s'^{n+1}},$$

and analogously the $u$-channel integral is finite for $t < \Delta_\phi / 2$, for $n$ specified above. This essentially leads to eq.(5.2) being the appropriate dispersion relation for $t < \Delta_\phi / 2$. This is another way of deriving our conclusions stated in section 4.1.3.

## 6   Discussion

In this paper, we have derived Froissart-like bounds for CFT Mellin amplitudes. We have seen that the flat space limit led to very interesting results for the flat space Froissart bounds. In particular, for 4 dimensional flat space the coefficient in front of the bound worked out to be $\pi / \rho m^2$ where using the map, $m$ was the mass of the external particle. We found that we could set $\rho = 1$. Hence, a naive comparison with experimental proton-proton data would give a better agreement than the original Froissart-Martin bound. There were key differences in other dimensions, the main one being that for $d > 6$, the number of subtractions could be greater than 2.

The physical implication of this last finding is not completely clear to us. We can venture a few guesses:

- It could be possible that the OPE asymptotics we used in the derivation need to be (discontinuously) different for $d \geq 5$, a possibility that does not appeal to us too much. However, we are not sure how and if this would resolve the difference. A power law asymptotics for the twist density will not suffice as we have argued earlier.

- It is a common folklore that there exist no interacting CFTs with a stress tensor in $d > 6$. The fact that we need more and more subtractions with increasing dimensionality may be tied in with this.

- In [37], the issue of consistent graviton S-matrices was considered. By demanding a polynomial bound of $s^2$ it was found that for flat spacetimes of dimensionality $(d + 1)$ greater than 6, there could be a six derivative polynomial that could be added, consistent with the bound. One cannot help wonder if there is a connection between our finding and theirs. Of course, for this, we would need to generalize our bounds to include external operators carrying spin. However, this does not seem insurmountable.

We should also emphasise the shortcomings of our derivation so that future work can remove them:

- Unlike Martin's rigorous derivation of the Martin ellipse, we assumed such an ellipse to exist. This is a strong assumption which one should examine carefully in the future, perhaps using the technology developed in [49]–we are at present investigating this.

- We restricted ourselves using OPE results necessary to reproduce the identity operator in the crossed channel, namely the MFT results. However, it is quite possible that the Tauberian type analysis in [17] will lead to stronger bounds. It will be very interesting to develop this further starting with eq.(4.37).

- As in the usual Froissart bound, our approach does not yet have anything to say in the situation with massless exchange is permitted. This appears to be a major shortcoming of the direction pursued in the present attempt and it will be important to consider avenues to remove this.

It will also be important to understand the connection with [21] in the future. Our goal was to come up with a framework that would in principle enable us to compute $1/R$ corrections to the standard Froissart bound and seems to be in a non-overlapping region of validity compared to [21]. In a forthcoming work [50], we will show that including the first $1/R$ correction, the bound (relevant for $3 + 1$ dimensional spacetime) takes the form

$$\bar{\mathcal{A}}(S) \leq \frac{\pi}{2\mu^2} S \ln^2 \frac{S}{S_0} - \left(\frac{c}{R^2\mu^2}\right)\left(\frac{S}{\mu^2}\right) \ln^4 \frac{S}{S_0}, \tag{6.1}$$

where $c > 0$ and hence the correction is negative (the correction appears to be negative in any dimensions). This seems to indicate that negatively curved spacetime would allow for less scattering than flat space, a result that does not appear to have been discussed at all in the literature. Presumably, for de Sitter space (naively $R \to i/H$, $H$ being the Hubble constant), the correction would be positive.

On the technical side, there is progress to be made. Ideally, we would need a better handle on the Mack polynomials, generalizing the bounds we used in this paper. The development of such technology would also be vital to probe $1/R$ corrections systematically to the Froissart$_{AdS}$ bounds considered in this paper–some results have been obtained in [50]. What such bounds have to say about the correlator in position space will also be of interest on the CFT side (see for instance [39, 40]).

Another line of questioning to ponder about is this. String theory suggests that there are extra compact dimensions. However, our finding was consistent with the $AdS_{d+1}/CFT_d$ correspondence. Is there any signature of the extra compact dimensions? Recently, it was pointed out [38], that the existence of extra dimensions can be probed perturbatively at one loop where new operators, other than the MFT operators, come in to the picture having different large $\Delta$ asymptotics. The growth of large extra dimensions (where the compact space is as large as the AdS radius) needs additional global symmetries. The spectral density in such a situation gets modified at one loop. However, our analysis has been nonperturbative and it is not clear to us what a nonperturbative version of this argument would be.

Finally, the issue of a consistent QFT saturating the Froissart bound has been of some interest in the past. Heisenberg came up with a model for hadron scattering which saturated this bound. There has been AdS/CFT inspired work addressing similar questions correlating the bound with the development of a black hole horizon [41, 42]. We found that the MFT density led to exactly the Froissart bound in the flat space limit for four dimensional flat space. The correlation between this and the Heisenberg model could be instructive to pursue.

# Acknowledgments

We thank F. Alday, B. Ananthanarayan, A. Gadde, R. Godbole, R. Gopakumar, S. Minwalla and A. Zhiboedov for useful discussions. We thank S. Pal for correspondence and especially P. Dey for pointing out typos in v1. A.S. gratefully acknowledges University of Oxford and CERN for hospitality during the course of this work. A.S. acknowledges support from a DST Swarnajayanti Fellowship Award DST/SJF/PSA-01/2013-14 and from the Tata Trusts for a travel grant.

# A  Mack polynomials: Conventions and properties

In this appendix, we will show explictly that in the "flat space limit" the leading asymptotic of Mack polynomial is Gegegnabuer polynomial. For that we need the explicit form of the Mack polynomial. There exists varied representation of Mack polynomials [11, 30, 32]. The normalization that we deploy for our cause is given by ,

$$\widehat{\mathcal{P}}_{\tau,\ell}(s,t) = \sum_{n=0}^{\ell}\sum_{m=0}^{\ell-n}\mu_{m,n}^{(\ell)}\left(\tau-s-\frac{\Delta_\phi}{2}\right)_m\left(-t+\frac{\Delta_\phi}{2}\right)_n, \tag{A.1}$$

with

$$\mu_{m,n}^{(\ell)} = 2^{-\ell}(-1)^{m+n}\binom{\ell}{m,n}$$
$$\times (\tau+\ell-m)_m(\tau+n)_{\ell-n}(\tau+m+n)_{\ell-m-n}(\ell+h-1)_{-m}(2\tau+2\ell-1)_{n-\ell}$$
$$\times \ _4F_3\left[\begin{array}{c}-m,\ 1-h+\tau,\ 1-h+\tau,\ n-1+2\tau+\ell\\ \tau+\ell-m,\ \tau+n,\ 2-2h+2\tau\end{array}\Big|1\right]. \tag{A.2}$$

Now we introduce the variable,

$$x = 1 + \frac{2t}{s-\Delta_\phi/2}. \tag{A.3}$$

Using this variable we rewrite the Mack polynomial as a function of $(s,x)$,

$$\widehat{\mathcal{P}}_{\tau,\ell}(s,x) = \sum_{n=0}^{\ell}\sum_{m=0}^{\ell-n}\mu_{m,n}^{(\ell)}\left(\tau-\left(s+\frac{\Delta_\phi}{2}\right)\right)_m\left(\frac{1-x}{2}(s-\Delta_\phi/2)+\frac{\Delta_\phi}{2}\right)_n. \tag{A.4}$$

Next we move on to giving the prescription for flat space limit. In the "flat space limit" we will consider,

$$s \gg \tau \gg 1. \tag{A.5}$$

The reason for this is that in our analysis, the twist sum lies between $\Delta_\phi$ and $\sqrt{2\Delta_\phi s}$ and since $s \gg \Delta_\phi/2$, the above consideration follows. In this limit we have for the leading asymptotic,

$$\left(\tau-s-\frac{\Delta_\phi}{2}\right)_m\left(\frac{1-x}{2}(s-\Delta_\phi/2)+\frac{\Delta_\phi}{2}\right)_n \sim (-1)^m s^{m+n}\left(\frac{1-x}{2}+\frac{\Delta_\phi}{(2s-\Delta_\phi)}\right)^n. \tag{A.6}$$

Clearly in the limit $s \gg 1$ the leading contribution in eq.(A.4) comes from $m = \ell-n$ so that we have,

$$\widehat{\mathcal{P}}_{\tau,\ell}(s,x) \sim s^\ell\sum_{n=0}^{\ell}(-1)^{\ell-n}\mu_{\ell-n,n}^{(\ell)}\left(\frac{1-x}{2}+\frac{\Delta_\phi}{(2s-\Delta_\phi)}\right)^n. \tag{A.7}$$

Now we will focus upon the $\tau \to \infty$ asymptotic of $(-1)^{\ell-n}\mu_{\ell-n,n}^{(\ell)}$. To start with, the leading large $\tau$ asymptotic of the factor premultiplying the hypergeometric function in eq.(A.2) is given by

$$2^{-2\ell+n}\tau^{\ell-n}\frac{(-\ell)_n}{n!}(\ell+h-1)_{n-\ell}, \tag{A.8}$$

where we have used the relation

$$(-1)^n\binom{\ell}{n} = \frac{(-\ell)_n}{n!}. \tag{A.9}$$

Next we focus upon the hypergeometric function above. Note that the $_4F_3$ above is balanced. Therefore we can use the following transformation due to Whipple to convert one balanced $_4F_3$ into another balanced $_4F_3$,

$$_4F_3\left[\begin{matrix} -p,\ a,\ b,\ c \\ d,\ e, f \end{matrix}\middle|1\right] = \frac{(e-a)_p(f-a)_p}{(e)_p(f)_p}\ _4F_3\left[\begin{matrix} -p,\ a,\ d-b,\ d-c \\ d,\ a+1-p-e, a+1-p-f \end{matrix}\middle|1\right]. \quad (A.10)$$

Using this we convert the $_4F_3$ in eq.(A.2) into ,

$$\frac{(h+n-1)_{\ell-n}(-h+\tau+1)_{\ell-n}}{(n+\tau)_{\ell-n}(2\tau-2h+2)_{\ell-n}}\ _4F_3\left[\begin{matrix} -(\ell-n),\ h+n-1,\ 1-\ell-\tau,\ 1-h+\tau \\ 2-\ell-h,\ n+\tau, n+h-\tau-\ell \end{matrix}\middle|1\right]. \quad (A.11)$$

Next we consider the limit $\tau \to \infty$ keeping $\ell$ fixed. The leading asymptotic is given by,

$$2^{n-\ell}\frac{(h+n-1)_{\ell-n}}{\tau^{\ell-n}}\ _2F_1\left[\begin{matrix} -(\ell-n),\ h+n-1 \\ 2-\ell-h \end{matrix}\middle|1\right]. \quad (A.12)$$

Thus clubbing together eq.(A.12) and eq.(A.8) we have,

$$(-1)^{\ell-n}\mu_{\ell-n,n}^{(\ell)} \sim 8^{-\ell}2^{2n}\frac{(-\ell)_n}{n!}\ _2F_1\left[\begin{matrix} -(\ell-n),\ h+n-1 \\ 2-\ell-h \end{matrix}\middle|1\right]. \quad (A.13)$$

Next using Chu-Vandermonde identity

$$_2F_1\left[\begin{matrix} -p,\ a \\ c \end{matrix}\middle|1\right] = \frac{(c-a)_p}{(c)_p}, \quad p \in \mathbb{Z}\backslash\mathbb{Z}^- \quad (A.14)$$

to obtain further

$$\begin{aligned}_2F_1\left[\begin{matrix} -(\ell-n),\ h+n-1 \\ 2-\ell-h \end{matrix}\middle|1\right] &= \frac{(3-\ell-2h-n)_{\ell-n}}{(2-\ell-h)_{\ell-n}} \\ &= \frac{\Gamma(2h-2+\ell+n)}{\Gamma(2h-2+2n)} \times \frac{\Gamma(h-1+n)}{\Gamma(h-1+\ell)} \\ &= \frac{(2h-2)_\ell}{(h-1)_\ell} \times \frac{(h-1)_n(2h-2+\ell)_n}{(2h-2)_{2n}}.\end{aligned} \quad (A.15)$$

Further using the identity,

$$\left(x+\frac{1}{2}\right)_n = 2^{2n}\frac{(2x)_{2n}}{(x)_n}, \quad n \in \mathbb{Z}\backslash\mathbb{Z}^- \quad (A.16)$$

we reach at,

$$2^{2n}\ _2F_1\left[\begin{matrix} -(\ell-n),\ h+n-1 \\ 2-\ell-h \end{matrix}\middle|1\right] = \frac{(2h-2)_\ell}{(h-1)_\ell} \times \frac{(2h-2+\ell)_n}{\left(h-\frac{1}{2}\right)_n}. \quad (A.17)$$

Thus collecting everything,

$$(-1)^{\ell-n}\mu_{\ell-n,n}^{(\ell)} \sim \frac{8^{-\ell}}{(h-1)_\ell}(2h-2)_\ell\frac{(-\ell)_n(\ell+2h-2)_n}{n!\left(h-\frac{1}{2}\right)_n}. \quad (A.18)$$

Putting this into eq.(A.7) we have in the limit $s \gg \tau \gg 1$, $s \gg \Delta_\phi$, with $x$ fixed,

$$\begin{aligned}\widehat{\mathcal{P}}_{\tau,\ell}(s,x) &\sim \frac{s^\ell}{8^\ell(h-1)_\ell}(2h-2)_\ell\sum_{n=0}^{\ell}\frac{(-\ell)_n(\ell+2h-2)_n}{n!\left(h-\frac{1}{2}\right)_n}\left(\frac{1-x}{2}+\frac{\Delta_\phi}{(2s-\Delta_\phi)}\right)^n \\ &= \frac{s^\ell}{8^\ell(h-1)_\ell}(2h-2)_\ell\ _2F_1\left[\begin{matrix} -\ell,\ 2(h-1)+\ell \\ (h-1)+\frac{1}{2} \end{matrix}\middle|\frac{1-x}{2}+\frac{\Delta_\phi}{(2s-\Delta_\phi)}\right] \\ &= \frac{s^\ell}{n_\ell}C_\ell^{(h-1)}(x)+O(s^{\ell-1}),\end{aligned} \quad (A.19)$$

where

$$n_\ell = 8^\ell \frac{(h-1)_\ell}{\ell!}. \tag{A.20}$$

Note that even for $x = 1$ the above expression holds true in the large $s$ limit. Thus we have the final asymptotic equivalence,

$$\boxed{\widehat{\mathcal{P}}_{\tau,\ell}(s, x) \sim \left(\frac{s}{8}\right)^\ell \frac{\ell!}{(h-1)_\ell} C_\ell^{(h-1)}(x), \quad s \gg \tau \gg 1.} \tag{A.21}$$

Some subleading corrections in a nice form to this are known and will appear in [50].

## B   Bounds on Gegenbauer polynomial

In this appendix we provide with a proof of the bounding relation eq.(4.29). The proof follows that given in [9]. We start with the following integral representation of the Gegenbauer polynomial $C_\ell^{(\alpha)}(x)$ for $x > 1$ (see for example [44]),

$$C_\ell^{(\alpha)}(x) = \frac{\Gamma(2\alpha + \ell)}{2^{2\alpha-1}\Gamma(\ell+1)\Gamma^2(\alpha)} \int_0^\pi \left[x + \sqrt{x^2 - 1}\cos\varphi\right]^\ell \sin^{2\alpha-1}\varphi \, d\varphi \tag{B.1}$$

$$= C_\ell^{(\alpha)}(1) \frac{\Gamma(2\alpha)}{2^{2\alpha-1}\Gamma^2(\alpha)} \int_0^\pi \left[x + \sqrt{x^2 - 1}\cos\varphi\right]^\ell \sin^{2\alpha-1}\varphi \, d\varphi. \tag{B.2}$$

Then we have the following integral representation for the normalized Gegenbauer polynomial [c.f. eq.(4.24)],

$$\mathbf{C}_\ell^{(\alpha)}(x) = \frac{\Gamma(2\alpha)}{2^{2\alpha-1}\Gamma^2(\alpha)} \int_0^\pi \left[x + \sqrt{x^2 - 1}\cos\varphi\right]^\ell \sin^{2\alpha-1}\varphi \, d\varphi. \tag{B.3}$$

Introducing the variable,

$$y = \frac{\sqrt{x^2 - 1}}{x}, \quad x > 1 \tag{B.4}$$

define,

$$H_\ell(y, \varphi_0, \varphi) := \frac{(1 + y\cos\varphi)^\ell}{(1 + y\cos\varphi_0)^\ell}, \tag{B.5}$$
$$G_\ell(y, \varphi_0) := \int_0^\pi H_\ell(y, \varphi_0, \varphi)(\sin\varphi)^{2\alpha-1}d\varphi.$$

Since $H_\ell(y, \varphi_0, \varphi)$ is an increasing function of $y$ for $0 < \varphi < \varphi_0 < \pi$ and decreasing function of $y$ for $0 < \varphi_0 < \varphi < \pi$, we can obtain the following inequality quite easily,

$$G_\ell(y, \varphi_0) \geq K(\varphi_0), \tag{B.6}$$

where

$$K(\varphi_0) = \int_0^{\varphi_0} d\varphi \, (\sin\varphi)^{2\alpha-1}. \tag{B.7}$$

Thus using eq.(B.6) we can obtain quite straightforwardly,

$$\mathbf{C}_\ell^{(\alpha)}(x) \geq \frac{\Gamma(2\alpha)}{2^{2\alpha-1}\Gamma^2(\alpha)} K(\varphi_0) \left[x + \sqrt{x^2 - 1}\cos\varphi_0\right]^\ell. \tag{B.8}$$

# C   A Derivation of Froissart-Martin Bound

In this appendix we provide with a derivation of the Froissart-Martin bound for the total scattering cross-section in usual 3+1 dimensional Minkowski spacetime. It is straightforward to generalize this derivation to any spacetime dimension. We consider $2 \to 2$ scattering of identical massive (mass being $m$) scalar spinless particles. Following [19], we present with the derivation of the Froissart-Martin bound for the averaged total scattering cross-section given by,

$$\bar{\sigma}(S) = \frac{1}{S-4m^2} \int_{4m^2}^{S} dS'(S'-4m^2)\sigma_{\text{tot}}(S'). \tag{C.1}$$

The Mandelstam variables $S, T, U$ are defined in 2.1.

First we write the S-Matrix $\mathcal{S}$ in the form $\mathcal{S} = 1 + i\mathcal{T}$. $\mathcal{T}$ is the scattering amplitude. Next, consider the partial wave expansion of the scattering amplitude,

$$\mathcal{T}(S,T) = \sqrt{\frac{S}{S-4m^2}} \sum_{\substack{\ell=0 \\ \ell\,\text{even}}}^{\infty} (2\ell+1)f_\ell(S)P_\ell\left(1+\frac{2T}{S-4m^2}\right), \tag{C.2}$$

where, $P_\ell(x)$ is the usual Legendre Polynomial. Now, we consider the absorptive part of the scattering amplitude given by,

$$A_S(S,T) = \text{Im}._S[\mathcal{T}(S,T)] = \sqrt{\frac{S}{S-4m^2}} \sum_{\substack{\ell=0 \\ \ell\,\text{even}}}^{\infty} (2\ell+1)\eta_\ell(S)P_\ell\left(1+\frac{2T}{S-4m^2}\right), \tag{C.3}$$

where

$$\text{Im}._S[f(s)] = \lim_{\epsilon\to 0} \frac{f(S+i\epsilon)-f(S-i\epsilon)}{2i} \tag{C.4}$$

and $\eta_\ell(S) = \text{Im}.[f_\ell(S)]$. Now by Optical theorem, the total scattering cross-section is related to $A(S, T=0)$. In fact, the total scattering cross-section is expressible in terms of $\{\eta_\ell(S)\}$,

$$\sigma_{\text{tot}}(S) = \frac{16\pi}{S-4m^2} \sum_{\substack{\ell=0 \\ \ell\,\text{even}}}^{\infty} (2\ell+1)\eta_\ell(S). \tag{C.5}$$

Therefore, we have for the averaged total scattering cross-section,

$$\bar{\sigma}(S) = \frac{16\pi}{S-4m^2} \int_{4m^2}^{S} dS' \left[\sum_{\substack{\ell=0 \\ \ell\,\text{even}}}^{\infty} (2\ell+1)\eta_\ell(S')\right]. \tag{C.6}$$

Unitarity of the S-Matrix, $\mathcal{S}^\dagger\mathcal{S} = \mathcal{S}\mathcal{S}^\dagger = 1$, gives the *partial wave unitarity constraint*[15],

$$0 \le \eta_\ell(S) \le 1 \ , \forall \ \ell \ge 0. \tag{C.7}$$

Next, consider the polynomial boundedness condition on the absorptive part [27, 29], that there exists a positive integer $n$ such that the integral quantity,

$$a_n := \int_{4m^2}^{\infty} \frac{d\bar{S}}{\bar{S}^{n+1}} A(\bar{S}, T=4m^2) \tag{C.8}$$

exists finitely.

---

[15]See, for example, [47] for a derivation.

Next, using the positivity of the Legendre polynomials $P_\ell(x)$ for $x > 1$ and the property that $P_\ell(x)$ is a strictly monotonically increasing fucntion of $\ell$ for $x > 1$, we reach the following inequality, after some straightforward algebra,

$$a_n \geq \sqrt{\frac{S}{S - 4m^2}} \, S^{-n-1} P_{L+2}\left(1 + \frac{8m^2}{S - 4m^2}\right) \sum_{\ell=L+2}^{\infty} (2\ell + 1) \int_{4m^2}^{S} d\bar{S} \, \eta_\ell(\bar{S}) \qquad (\text{C.9})$$

for some $L > 0$. Next we shift our attention to $\bar{\sigma}(S)$. We write the the same as

$$\bar{\sigma}(S) = \frac{16\pi}{S - 4m^2} \sum_{\substack{\ell=0 \\ \ell \text{ even}}}^{L} (2\ell + 1) \int_{4m^2}^{S} d\bar{S} \, \eta_\ell(\bar{S}) + \frac{16\pi}{S - 4m^2} \sum_{\ell=L+2}^{\infty} (2\ell + 1) \int_{4m^2}^{S} d\bar{S} \, \eta_\ell(\bar{S}). \quad (\text{C.10})$$

Next, using eq.(C.7) and eq.(C.9) one obtains quite straightforwardly,

$$\bar{\sigma}(S) \leq \frac{16\pi}{S - 4m^2}\left[(S - 4m^2)\frac{1}{2}(L + 1)(L + 2) + \sqrt{\frac{S - 4m^2}{S}} \frac{S^{n+1}}{P_{L+2}\left(1 + \frac{8m^2}{S - 4m^2}\right)} a_n\right]. \quad (\text{C.11})$$

Let us now focus on the second term above. First we note the following inequality satisfied by the Legendre polynomial,

$$P_\ell(x) \geq \frac{\varphi_0}{\pi}\left(x + \cos\varphi_0 \sqrt{x^2 - 1}\right)^\ell \quad x > 1, \ 0 < \varphi_0 < \pi. \qquad (\text{C.12})$$

This readily follows from putting $\alpha = 1/2$ into eq.(B.8). Now using this, we have in the limit $L \gg 1$ and $S \gg 4m^2$,

$$\sqrt{\frac{S - 4m^2}{S}} \frac{S^{n+1}}{P_{L+2}\left(1 + \frac{8m^2}{S - 4m^2}\right)} a_n \approx \frac{S^{n+1}}{P_L\left(1 + \frac{8m^2}{S}\right)} a_n \lesssim \frac{\pi a_n}{\varphi_0} S^{n+1}\left[1 + \cos\varphi_0 \frac{4m}{\sqrt{S}}\right]^{-L}. \quad (\text{C.13})$$

Using the above inequality into eq.(C.11), we obtain the following inequality,

$$\bar{\sigma}(S) \lesssim 16\pi\left[\frac{L^2}{2} + \frac{\pi a_n}{\varphi_0} S^n\left(1 + \cos\varphi_0 \frac{4m}{\sqrt{S}}\right)^{-L}\right] = 8\pi L^2\left[1 + \frac{2\pi a_n}{\varphi_0} \frac{S^n}{L^2}\left(1 + \cos\varphi_0 \frac{4m}{\sqrt{S}}\right)^{-L}\right]. \quad (\text{C.14})$$

The $\ell$ sum gets truncated effectively if the second term inside the parenthesis above is very small compared to 1. Let us consider the marginal case when this is equal to 1. The $L$ determined by this marginal case is the one where we will decide to truncate the $\ell$-sum effectively.

Considering an ansatz for the $L$ of the form

$$L = A_0 \, s^a \ln^b s \qquad (\text{C.15})$$

one readily obtains, in the limit $S \gg 4m^2$, that in the leading order in $S$, the marginal $L$ is given by,

$$a = 1/2, \ b = 1 \, , in A_0 = \frac{n - 1}{4m \cos\varphi_0}. \qquad (\text{C.16})$$

Thus the marginal $L$−cutoff is given by,

$$L = \frac{(n - 1)}{\cos\varphi_0}\sqrt{\frac{S}{16m^2}} \ln S. \qquad (\text{C.17})$$

Now, truncating the $\ell$-sum contributing to $\bar{\sigma}(S)$ to $L$ we obtain,

$$\bar{\sigma}(S) \leq \frac{\pi}{2m^2}(n-1)^2 S \ln^2 \frac{S}{S_0}. \tag{C.18}$$

where $S_0$ is some scale to make the argument of the ln diemsnionless. Further, $n$ can be fixed to 2 using $Phragmen - Lindelof$ theorem giving finally

$$\bar{\sigma}(S) \leq \frac{\pi}{2m^2} S \ln^2 \frac{S}{S_0}. \tag{C.19}$$

# D  The Conformal Partial Wave Expansion in Mellin Space

In this appendix, we give a short account of the conformal partial wave expansion in Mellin space, leading to eq.(3.7). First, consider the usual conformal partial wave expansion in position space. For a 4−point correlator of identical scalar primaries of conformal dimension $\Delta_\phi$, the $s$−channel conformal partial wave expansion is given by,

$$\mathcal{G}(u,v) = \sum_{\tau,\ell} C_{\tau,\ell}\, G_{\tau,\ell}(u,v), \tag{D.1}$$

where $u, v, \mathcal{G}(u,v)$ are defined in eq.(3.3) and $\tau = (\Delta - \ell)/2$ with $\Delta, \ell$ being respectively the scaling dimension and spin of the exchanged primary and $C_{\tau,\ell}$ is the corresponding OPE coefficient squared. We are considering unitary theories where $C_{\tau,\ell} \geq 0$. Here, $G_{\tau,\ell}(u,v)$ is the $s$−channel conformal block with the normalization that, in the limit $v \ll u \ll 1$ [45] the conformal block has the asymptotic form

$$G_{\tau,\ell}(u,v) \sim u^\tau f_{\tau,\ell}(v). \tag{D.2}$$

With our definition for the Mellin amplitude, eq.(3.5), the $\mathcal{M}(s,t)$ admits a partial wave expansion [32]

$$\mathcal{M}(s,t) = \sum_{\tau,\ell} C_{\tau,\ell} \widehat{\mathcal{N}}_{\tau,\ell} \frac{\sin^2 \pi\left(\frac{\Delta_\phi}{2}-s\right)}{\sin^2 \pi\left(\Delta_\phi - \tau - \frac{\ell}{2}\right)} \frac{\Gamma\left(\tau-s-\frac{\Delta_\phi}{2}\right)\Gamma\left(h-\tau-\ell-\frac{\Delta_\phi}{2}-s\right)}{\Gamma^2\left(\frac{\Delta_\phi}{2}-s\right)} \widehat{\mathcal{P}}_{\tau,\ell}(s,t), \tag{D.3}$$

with

$$\widehat{\mathcal{N}}_{\tau,\ell} = 2^\ell \frac{(2\tau+2\ell-1)\Gamma^2(2\tau+2\ell-1)}{\Gamma(2\tau+\ell-1)\Gamma(h-2\tau-\ell)\Gamma^4(\tau+\ell)}. \tag{D.4}$$

This expansion above is just the Mellin space version of the $s$−channel conformal block expansion eq.(D.1) above with the normalization eq.(D.2). Now we will massage this expression into eq.(3.5).

The starting point is the observation that the Euler-beta function $B(x,y) = \frac{\Gamma(x)\Gamma(y)}{\Gamma(x+y)}$ admits the following expansion,

$$B(x,y) = \sum_{n=0}^{\infty} \frac{(-1)^n (y-n)_n}{n!(x+n)} = \sum_{n=0}^{\infty} \frac{(-1)^n (x-n)_n}{n!(y+n)}. \tag{D.5}$$

In other words, the Euler-beta function can be expanded in terms of the poles of either of the Gamma functions. This is not true for the usual Gamma function which needs, in addition,

a regular piece for the expansion to be valid. Using this, we will massage the $s-$dependent combination of Gamma functions appearing in eq.(D.3) into the form,

$$\frac{\Gamma\left(\tau-s-\frac{\Delta_\phi}{2}\right)\Gamma\left(h-\tau-\ell-\frac{\Delta_\phi}{2}-s\right)}{\Gamma^2\left(\frac{\Delta_\phi}{2}-s\right)} = B\left(\tau-s\frac{\Delta_\phi}{2},\Delta_\phi-\tau\right)\frac{\Gamma\left(h-\tau-\ell-s-\frac{\Delta_\phi}{2}\right)}{\Gamma\left(\frac{\Delta_\phi}{2}-s\right)\Gamma(\Delta_\phi-\tau)}. \quad \text{(D.6)}$$

Next, we use eq.(D.5) to expand the beta function leading to[16]

$$\begin{aligned}
\mathcal{M}(s,t) &= \sum_{\tau,\ell}\mathcal{F}_{\tau,\ell}(s)\sum_{n=0}^{\infty}\frac{(-1)^n(\Delta_\phi-\tau-n)_n}{n!(\tau-s-\Delta_\phi/2+n)}\frac{\Gamma(h-\tau-\ell-s-\Delta_\phi/2)}{\Gamma(\Delta_\phi/2-s)\Gamma(\Delta_\phi-\tau)}\widehat{\mathcal{P}}_{\tau,\ell}(s,t) \\
&= \sum_{\tau,\ell}\mathcal{F}_{\tau,\ell}(s)\sum_{n=0}^{\infty}\frac{(-1)^n(\Delta_\phi-\tau-n)_n}{n!(\tau-s-\Delta_\phi/2+n)}\frac{\Gamma(h-2\tau-\ell-n)}{\Gamma(\Delta_\phi-\tau-n)\Gamma(\Delta_\phi-\tau)}\widehat{\mathcal{P}}_{\tau,\ell}(s,t)+\ldots \\
&= \sum_{\tau,\ell}\mathcal{F}_{\tau,\ell}(s)\frac{\Gamma(h-2\tau-\ell)\widehat{\mathcal{P}}_{\tau,\ell}}{\Gamma^2(\Delta_\phi-\tau)(\tau-s-\Delta_\phi/2)} \\
&\quad \times \; {}_3F_2\left[\begin{matrix}\tau-s-\frac{\Delta_\phi}{2},1+\tau-\Delta_\phi,1+\tau-\Delta_\phi \\ 1+\tau-s-\frac{\Delta_\phi}{2},2\tau+\ell-h+1\end{matrix}\,\middle|\,1\right]+\ldots
\end{aligned}$$

where in the second line we have Taylor expanded the ratio of Gamma functions around the $s-$pole $s = \tau-\frac{\Delta_\phi}{2}+n$. The dots represent regular terms and we assume that, this final form will give the same residues at the location of the physical poles as eq.(D.3). However, notice that ${}_3F_2$ form now is devoid of the zeros coming from the inverse $\Gamma^2\left(\frac{\Delta_\phi}{2}-s\right)$ factor and differs from the form in eq.(D.3) by regular terms hidden in the dots. We assume that, these regular terms converge so that they will not contribute to the absorptive part of $\mathcal{M}(s,t)$. Thus we can consider this form of $\mathcal{M}(s,t)$ modulo the regular terms. Finally restoring all the factors we reach the desired form eq.(3.7). Again to emphasise, we could have started with the Dolan-Osborn form in eq.(D.3) and obtained the imaginary part directly from there as well–the results would be identical.

# E  Asymptotic analysis of the sum eq.(4.37)

The summand of the $\tau-$ sum is given by eq.(4.38). Barring all the prefactors, we will concentrate upon the following sum,

$$\sum_{\tau=\Delta_\phi}^{\tau_\star}\tau^{4-2h}\sin^2[\pi(\tau-\Delta_\phi)], \quad \tau_\star = \sqrt{(2\Delta_\phi+\ell)s}. \quad \text{(E.1)}$$

We need to be able to this sum. Generally we can resort to integrals assuming that the $\tau-$spectrum can be considered to be continuous so that we can resort to the integral. However this is actually not true. But for the purpose of the asymptotic evaluations we can resort to integral. Thus we are interested in the integral,

$$\int_{\Delta_\phi}^{\tau_\star}d\tau\;\tau^{4-2h}\sin^2[\pi(\tau-\Delta_\phi)]. \quad \text{(E.2)}$$

---

[16]Here we have defined $\mathcal{F}_{\tau,\ell}(s) := C_{\tau,\ell}\,\widehat{\mathcal{N}}_{\tau,\ell}\,\frac{\sin^2\pi\left(\frac{\Delta_\phi}{2}-s\right)}{\sin^2\pi\left(\Delta_\phi-\tau-\frac{\ell}{2}\right)}$ to avoid cumbersome notation.

We can start with the indefinite version of the above integral,

$$
F(\tau) := \int d\tau \; \tau^{4-2h} \sin^2[\pi(\tau - \Delta_\phi)]
$$
$$
= \frac{e^{-2i\pi\Delta_\phi}}{4(4(h-3)h+5)} \Big[ (2h-1)\tau^{5-2h}
$$
$$
\Big[ -2e^{2i\pi\Delta_\phi} + (2h-5) \big\{ e^{4i\pi\Delta_\phi} E_{2h-4}(2i\pi\tau) + E_{2h-4}(-2i\pi\tau) \big\} \Big] + 8e^{2i\pi\Delta_\phi}\Delta_\phi^{5-2h} \Big].
$$
(E.3)

Now we will consider two asymptotic limits. First we will consider the asymptotic of $F(\Delta_\phi)$ in the limit of large $\Delta_\phi$. This limit is given by,

$$
F(\Delta_\phi) \sim \frac{\Delta_\phi^{5-2h}}{10-4h}.
$$
(E.4)

Also in the limit $\tau_\star \gg 1$,

$$
F(\tau_\star) \sim \frac{\tau_\star^{5-2h}}{10-4h}.
$$
(E.5)

Note that since we are considering $s \gg \Delta_\phi$ hence for $h > \frac{5}{2}$, $F(\Delta_\phi)$ dominates over $F(\tau_\star)$. Thus for $h < 5/2$ we can write,

$$
\int_{\Delta_\phi}^{\tau_\star} d\tau \; \tau^{4-2h} \sin^2[\pi(\tau - \Delta_\phi)] \sim \frac{\tau_\star^{5-2h}}{10-4h} = \frac{1}{10-4h}\Big[ s(2\Delta_\phi + \ell) \Big]^{\frac{5}{2}-h}, \quad \tau_\star \to \infty
$$
(E.6)

and for $h > 5/2$ we can use,

$$
\int_{\Delta_\phi}^{\tau_\star} d\tau \; \tau^{4-2h} \sin^2[\pi(\tau - \Delta_\phi)] \sim \frac{\Delta_\phi^{5-2h}}{4h-10}, \quad \Delta_\phi \to \infty.
$$
(E.7)

Now we would like to consider the case $h = 5/2$. Note that we can not put $h = 5/2$ directly into the either asymptotic expressions that we have obtained so far, eq.(E.6) or eq.(E.7), because the denominator vanishes identically thus resulting into a non-removable singular structure. This is however expected because on putting $h = 5/2$ into eq.(E.2) we see that the integrand being $\sim \frac{1}{\tau}$ has a logarithmic singularity. Thus to tackle this case we will start with the integral eq.(E.2) and put $h = 5/2$ into it so that the integral we need to do is,

$$
\int_{\Delta_\phi}^{\tau_\star} \frac{d\tau}{\tau} \sin^2[\pi(\tau - \Delta_\phi)].
$$
(E.8)

Therefore,

$$
F(\tau)\Big|_{h=\frac{5}{2}} = \frac{1}{2}\Big[ \log(\pi\tau) - \text{Ci}(2\pi\tau)(-\cos(2\pi\Delta_\phi)) - \sin(2\pi\Delta_\phi)\text{Si}(2\pi\tau) \Big].
$$
(E.9)

Clearly we can write the asymptotic expression,

$$
F(\tau)\Big|_{h=\frac{5}{2}} \sim \frac{1}{2}\log(\tau), \quad \tau \to \infty.
$$
(E.10)

Thus in the limit $s \gg \Delta_\phi \gg 1$ and also $s \gg \ell$, we can write the leading order asymptotic expression,

$$
\int_{\Delta_\phi}^{\tau_\star} \frac{d\tau}{\tau} \sin^2[\pi(\tau - \Delta_\phi)] \sim \frac{1}{4}\log s.
$$
(E.11)

# F  Asymptotic evaluations of various sums

In this appendix we provide certain asymptotes of summation expressions.

1) We come across the following sum in various occasions of our analysis,

$$\sum_{\substack{\ell \\ \ell \text{ even}}}^{L} \ell^{2h-2}. \tag{F.1}$$

We are mostly interested in the large $L$ asymptotic of the above sum. To obtain so, first we we have

$$\sum_{\substack{\ell \\ \ell \text{ even}}}^{L} \ell^{\alpha} = 2^{\alpha} H_{\frac{L}{2}}^{(-\alpha)}. \tag{F.2}$$

Next, considering the asymptotic of the $r^{th}$ order Harmonic number

$$H_x^{(r)} \sim \frac{x^{1-r}}{1-r}, \quad x \to \infty \tag{F.3}$$

we can write,

$$\sum_{\substack{\ell \\ \ell \text{ even}}}^{L} \ell^{\alpha} \sim \frac{L^{1+\alpha}}{2+2\alpha}. \tag{F.4}$$

Thus we have finally,

$$\sum_{\substack{\ell \\ \ell \text{ even}}}^{L} \ell^{2h-2} \sim \frac{L^{2h-1}}{4h-2}. \tag{F.5}$$

2) Next, we consider the $\ell-$sum appearing in the eq.(4.43) and eq.(4.69). The sum is of the generic form,

$$\sum_{\substack{\ell \\ \ell \text{ even}}}^{L} \ell^{b} (a+\ell)^{c}, \quad a > 0; \ b, c \in \mathbb{R}. \tag{F.6}$$

Now for our purpose, we are generally interested in large $a$, large $L$ asymptotic of the sum. The case that interests us is the one where we consider $a \gg L$. To tackle the sum we can take help of the Euler-Maclaurin formula and to *leading order in L* we can replace the sum by integral,

$$\sum_{\substack{\ell \\ \ell \text{ even}}}^{L} \ell^{b} (a+\ell)^{c} \sim 2^{b} \int_{0}^{L/2} dx \, x^{b} (a+2x)^{c}. \tag{F.7}$$

This integral can be expressed in terms of incomplete Beta function as,

$$2^{b} \int_{0}^{L/2} dx \, x^{b} (a+2x)^{c} = \frac{1}{2} a^{c} L^{b+1} \Gamma(b+1) \, {}_2\tilde{F}_1\left(b+1, -c; b+2; -\frac{L}{a}\right), \tag{F.8}$$

where

$$ {}_2\tilde{F}_1(a, b, c; z) = \frac{{}_2F_1(a, b, c; z)}{\Gamma(c)}. \tag{F.9}$$

Now we will consider further $a \gg L \gg 1$. The desired asymptotic in this limit is given by,

$$2^b \int_0^{L/2} dx \; x^b \, (a + 2x)^c \sim a^c \frac{L^{b+1}}{2b+2} \tag{F.10}$$

so that we can write finally wrapping up everything,

$$\sum_{\substack{\ell \\ \ell \text{ even}}}^L \ell^b \, (a + \ell)^c \sim a^c \frac{L^{b+1}}{2b+2}. \tag{F.11}$$

As for the $\ell$−sum appearing in eq.(4.43) we put the values $a = 2\Delta_\phi$, $b = 2h - 2$ and $c = (3 - 2h)/2$ to obtain,

$$\sum_{\substack{\ell \\ \ell \text{ even}}}^L \ell^{2h-2} \, (2\Delta_\phi + \ell)^{\frac{3-2h}{2}} \sim (2\Delta_\phi)^{\frac{3-2h}{2}} \frac{L^{2h-1}}{4h-2}, \quad 2\Delta_\phi \gg L \gg 1. \tag{F.12}$$

Similarly for the $\ell$−sum appearing in eq.(4.69), one puts $a = 2\Delta_\phi$, $b = h-1$, $c = (3-2h)/2$ to obtain,

$$\sum_{\substack{\ell \\ \ell \text{ even}}}^L \ell^{h-1} \, (2\Delta_\phi + \ell)^{\frac{3-2h}{2}} \sim (2\Delta_\phi)^{\frac{3-2h}{2}} \frac{L^h}{2h}, \quad 2\Delta_\phi \gg L \gg 1. \tag{F.13}$$

# G   Considering the primaries only

We have seen that, in the flat space limit, the contribution towards the Mellin amplitude of a conformal family corresponding to a primary operator with dimension $\Delta$ is peaked not at the primary, rather at a descendant as dictated by eq.(4.3), eq.(4.4). Thus the natural expectation is that, if we consider just the contributions of the primaries then it should have vanishingly small contribution towards the full result in the flat space limit. It is a worthwhile exercise to look into this explicitly. In this section, we take up this job and and bound the primary contribution towards the Mellin amplitude.

We start with the following definition which isolate the contribution of the primaries towards $\mathcal{A}_M(s, t)$,

$$\mathcal{A}_M^{(p)}(s, t) := \sum_{\substack{\tau, \ell \\ \ell \text{ even}}} C_{\tau, \ell} \, \text{Im}[f_{\tau, \ell}(s)]^{(p)} \, \widehat{\mathcal{P}}_{\tau, \ell}(s, t), \tag{G.1}$$

with

$$\text{Im}[f_{\tau, \ell}(s)]^{(p)} := \pi \mathcal{N}_{\tau, \ell} \frac{\Gamma^2(\tau + \ell + \Delta_\phi - h)}{\Gamma(2\tau + \ell - h + 1)} \frac{\sin^2 \pi \left(\frac{\Delta_\phi}{2} - s\right)}{\sin^2 \pi \left(\Delta_\phi - \tau - \frac{\ell}{2}\right)} \delta(s + \Delta_\phi/2 - \tau). \tag{G.2}$$

With the help of this expression we can define $\bar{\mathcal{A}}_M^{(p)}(s)$ and $\mathfrak{a}_{n, \Delta_\phi/2}^{(p)}$. We have

$$\bar{\mathcal{A}}_M^{(p)}(s) = \frac{1}{s - \frac{\Delta_\phi}{2}} \int_{\Delta_\phi/2}^s ds' \mathcal{A}_M^{(p)}(s', t = 0),$$

$$= \frac{\pi}{s - \frac{\Delta_\phi}{2}} \int_{\Delta_\phi/2}^s ds' \sum_{\substack{\ell \\ \ell \text{ even}}} \sum_{\tau = h-1}^\infty C_{\tau, \ell} \mathcal{N}_{\tau, \ell} \frac{\Gamma^2(\tau + \ell + \Delta_\phi - h)}{\Gamma(2\tau + \ell - h + 1)} \frac{\sin^2 \pi \left(\frac{\Delta_\phi}{2} - s\right)}{\sin^2 \pi \left(\Delta_\phi - \tau - \frac{\ell}{2}\right)} \tag{G.3}$$

$$\times \delta(s + \Delta_\phi/2 - \tau) \widehat{\mathcal{P}}_{\tau, \ell}(s', 0).$$

Next we will do a small trick to handle the above quantity. We introduce an integration over a sum of delta functions in $x$ to rewrite,

$$\int_{\Delta_\phi/2}^{s} ds' \sum_{\substack{\ell \\ \ell \text{ even}}} \int_0^\infty dx \sum_{\tau=h-1}^\infty \delta(x-\tau) C_\ell(x) \mathcal{N}_\ell(x) \frac{\Gamma^2(x+\ell+\Delta_\phi-h)}{\Gamma(2x+\ell-h+1)} \frac{\sin^2 \pi\left(\frac{\Delta_\phi}{2}-s'\right)}{\sin^2 \pi\left(\Delta_\phi-x-\frac{\ell}{2}\right)}$$
$$\times \delta\left(s'+\frac{\Delta_\phi}{2}-x\right), \tag{G.4}$$

where

$$C_\ell(\tau) \equiv C_{\tau,\ell}, \quad \mathcal{N}_\ell(\tau) \equiv \mathcal{N}_{\tau,\ell}. \tag{G.5}$$

Now we will change the order of integration and do the $s'$ integral first. Note that the $s'$ dependent delta function will give nonzero contribution when,

$$s' = x - \frac{\Delta_\phi}{2}. \tag{G.6}$$

But further we have $\Delta_\phi < s' < s$ which gives us a condition on $x$,

$$\Delta_\phi \leq x \leq s + \frac{\Delta_\phi}{2}. \tag{G.7}$$

This condition effectively truncates the $\tau$ sum above and produces,

$$\sum_{\substack{\ell \\ \ell \text{ even}}} \sum_{\tau=\Delta_\phi}^{s+\frac{\Delta_\phi}{2}} C_{\tau,\ell} \, \mathcal{N}_{\tau,\ell} \frac{\Gamma^2(\tau+\ell+\Delta_\phi-h)}{\Gamma(2\tau+\ell-h+1)} \frac{\sin^2\left[\pi\left(\Delta_\phi-\tau\right)\right]}{\sin^2\left[\pi\left(\Delta_\phi-\tau-\frac{\ell}{2}\right)\right]} \widehat{\mathcal{P}}_{\tau,\ell}\left(\tau-\frac{\Delta_\phi}{2},0\right). \tag{G.8}$$

Next we would like to investigate the Mack polynomial $\widehat{\mathcal{P}}_{\tau,\ell}(\tau-\Delta_\phi/2,t)$. We have,

$$\widehat{\mathcal{P}}_{\tau,\ell}\left(\tau-\frac{\Delta_\phi}{2},t\right) = \sum_{n=0}^\ell \mu_{0,n}^{(\ell)} \left(\frac{\Delta_\phi}{2}-t\right)_n. \tag{G.9}$$

Further,

$$\mu_{0,n}^{(\ell)} = 2^{-\ell}(-1)^n \binom{\ell}{n}(\tau+n)_{\ell-n}^2 (2\tau+2\ell-1)_{n-\ell}$$
$$= 2^{-\ell}\frac{(-\ell)_n}{n!}\frac{\Gamma^2(\tau+\ell)}{\Gamma^2(\tau+n)} \times (2\tau+\ell-1)_n \times \frac{\Gamma(2\tau+\ell-1)}{\Gamma(2\tau+2\ell-1)}, \tag{G.10}$$
$$= 2^{-\ell}\frac{(\tau)_\ell^2}{(2\tau+\ell-1)_\ell} \times \frac{(-\ell)_n (2\tau+\ell-1)_n}{n! \, (\tau)_n^2}.$$

Thus we have,

$$\widehat{\mathcal{P}}_{\tau,\ell}\left(\tau-\frac{\Delta_\phi}{2},t\right) = 2^{-\ell}\frac{(\tau)_\ell^2}{(2\tau+\ell-1)_\ell} \, {}_3F_2\left[\begin{matrix} -\ell, \, 2\tau+\ell-1, \, \frac{\Delta_\phi}{2}-t \\ \tau, \, \tau \end{matrix} \middle| 1\right]. \tag{G.11}$$

Also we have,

$$A_{\tau,\ell} := \mathcal{N}_{\tau,\ell}\frac{\Gamma^2(\tau+\ell+\Delta_\phi-h)}{\Gamma(2\tau+\ell-h+1)} = 2^\ell \frac{(2\tau+2\ell-1)\Gamma^2(2\tau+2\ell-1)}{\Gamma(2\tau+\ell-1)\Gamma^4(\tau+\ell)\Gamma^2(\Delta_\phi-\tau)}. \tag{G.12}$$

Putting everything together we have,

$$\bar{\mathcal{A}}_M^{(p)}(s) = \frac{\pi}{s - \frac{\Delta_\phi}{2}} \sum_{\substack{\ell \\ \ell \text{ even}}} \sum_{\tau=\Delta_\phi}^{s+\frac{\Delta_\phi}{2}} C_{\tau,\ell} \frac{\Gamma(2\tau+2\ell)}{\Gamma^2(\tau)\Gamma^2(\tau+\ell)\Gamma^2(\Delta_\phi-\tau)} \,_3F_2\left[\begin{matrix} -\ell,\ 2\tau+\ell-1,\ \frac{\Delta_\phi}{2} \\ \tau,\ \tau \end{matrix}\bigg|1\right],$$

(G.13)

where we have used the fact of $\ell$ being even to get rid of the ratio of the sin squares. To bound this, we will have to "effectively cut" the $\ell$ sum to some finite summation. To determine that "$\ell$ cutoff" we will exploit the information of the polynomial boundedness of the Mellin amplitude that we have assumed. To do so we will take help of $\mathfrak{a}_{n,\Delta_\phi/2}$ as follows

$$\begin{aligned}
\mathfrak{a}_{n,\Delta_\phi/2}^{(p)} &= \int_{\frac{\Delta_\phi}{2}}^\infty \frac{d\bar{s}}{\bar{s}^{n+1}}\, \mathcal{A}_M^{(p)}(\bar{s}, t=\Delta_\phi/2) \\
&> \int_{\frac{\Delta_\phi}{2}}^s \frac{d\bar{s}}{\bar{s}^{n+1}}\, \mathcal{A}_M^{(p)}(\bar{s}, t=\Delta_\phi/2) > s^{-(n+1)} \int_{\frac{\Delta_\phi}{2}}^s ds'\, \mathcal{A}_M^{(p)}(s', t=\Delta_\phi/2).
\end{aligned}$$

(G.14)

The first inequality follows from the integrand being always positive[17] for unitary theories because for unitary theories $C_{\tau,\ell} \in \mathbb{R}^\geq$. The second inequality follows because $s^{-(n+1)}$ is a monotonically decreasing function of $s$ for $n > 0$. Next ploughing through the same steps as before we have,

$$\begin{aligned}
\mathfrak{a}_{n,\Delta_\phi/2}^{(p)}\, s^{n+1} &> \pi \sum_{\substack{\ell \\ \ell \text{ even}}} \sum_{\tau=\Delta_\phi}^{s+\frac{\Delta_\phi}{2}} C_{\tau,\ell} \frac{\Gamma(2\tau+2\ell)}{\Gamma^2(\tau)\Gamma^2(\tau+\ell)\Gamma^2(\Delta_\phi-\tau)} \\
&> \pi \sum_{\substack{\ell=L+2 \\ \ell \text{ even}}}^\infty \sum_{\tau=\Delta_\phi}^{s+\frac{\Delta_\phi}{2}} C_{\tau,\ell} \frac{\Gamma(2\tau+2\ell)}{\Gamma^2(\tau)\Gamma^2(\tau+\ell)\Gamma^2(\Delta_\phi-\tau)},
\end{aligned}$$

(G.15)

where the last inequality follows on using the positivity of the summand. Here $L$ is some $\ell$ value which is presumably large. This basically defines the *tail* of the series.

## G.1 Determining the $\ell$-cutoff

To make use of this above inequality eq.(G.15) in order to find out the "$\ell$ cutoff" as mentioned above we turn our attention to once again to eq.(G.13) and write the same as follows,

$$\bar{\mathcal{A}}_M^{(p)}(s) = \frac{\pi}{s - \frac{\Delta_\phi}{2}} \sum_{\substack{\ell=0 \\ \ell \text{ even}}}^L \sum_{\tau=\Delta_\phi}^{s+\frac{\Delta_\phi}{2}} C_{\tau,\ell} \frac{\Gamma(2\tau+2\ell)}{\Gamma^2(\tau)\Gamma^2(\tau+\ell)\Gamma^2(\Delta_\phi-\tau)} \,_3F_2\left[\begin{matrix} -\ell,\ 2\tau+\ell-1,\ \frac{\Delta_\phi}{2} \\ \tau,\ \tau \end{matrix}\bigg|1\right] + \Re(s),$$

(G.16)

with the "remainder" term being,

$$\Re(s) = \frac{\pi}{s - \frac{\Delta_\phi}{2}} \sum_{\substack{\ell=L+2 \\ \ell \text{ even}}}^\infty \sum_{\tau=\Delta_\phi}^{s+\frac{\Delta_\phi}{2}} C_{\tau,\ell} \frac{\Gamma(2\tau+2\ell)}{\Gamma^2(\tau)\Gamma^2(\tau+\ell)\Gamma^2(\Delta_\phi-\tau)} \,_3F_2\left[\begin{matrix} -\ell,\ 2\tau+\ell-1,\ \frac{\Delta_\phi}{2} \\ \tau,\ \tau \end{matrix}\bigg|1\right].$$

(G.17)

---

[17] the positivity is in the sense of distribution i.e, on integrating against a Schwartz function the sign of the function remains unaltered.

Next we will use certain properties of the $_3F_2$ polynomial appearing above. For future reference let us introduce the following defining notation

$$\widehat{Q}_\ell(\tau) = {}_3F_2\left[\begin{array}{c} -\ell, \ 2\tau+\ell-1, \ \frac{\Delta_\phi}{2} \\ \tau, \ \tau \end{array}\middle| 1\right].$$  (G.18)

This polynomial has two crucial properties that will come to our use to a great extent. These are the following,

I. The first useful property that we have is that $\widehat{Q}_\ell(\tau)$ is a decreasing function of $\ell$. The most general "observation"[18] is that this is true *separately* for even spins and odd spins. Using this property therefore we can write,

$$\Re(s) < \frac{\pi}{s-\frac{\Delta_\phi}{2}} \sum_{\substack{\ell=L+2 \\ \ell \text{ even}}}^{\infty} \sum_{\tau=\Delta_\phi}^{s+\frac{\Delta_\phi}{2}} C_{\tau,\ell} \frac{\Gamma(2\tau+2\ell)}{\Gamma^2(\tau)\Gamma^2(\tau+\ell)\Gamma^2(\Delta_\phi-\tau)} \widehat{Q}_{L+2}(\tau).$$  (G.19)

II. The second property that we will make use of is that generally for large enough $\tau$ one has $\widehat{Q}_\ell(\tau)$ an increasing function of $\tau$. Now the important part of this statement is *large enough $\tau$*. For practical reasons this is synonymous with $\tau \gg \Delta_\phi$ for our case. The reason for emphasizing this is that in general vary near to $\tau = \Delta_\phi$ the polynomial $\widehat{Q}_\ell(\tau)$ decreases for some time reaching a minimum and then once again starts increasing and maintains the increasing trend with increasing $\tau$. Since we are ultimately interested in $s \gg \Delta_\phi/2$ we can safely use this property of $\widehat{Q}_\ell(\tau)$ to write,

$$\Re(s) < \frac{\pi}{s-\frac{\Delta_\phi}{2}} \widehat{Q}_{L+2}\left(s+\frac{\Delta_\phi}{2}\right) \sum_{\substack{\ell=L+2 \\ \ell \text{ even}}}^{\infty} \sum_{\tau=\Delta_\phi}^{s+\frac{\Delta_\phi}{2}} C_{\tau,\ell} \frac{\Gamma(2\tau+2\ell)}{\Gamma^2(\tau)\Gamma^2(\tau+\ell)\Gamma^2(\Delta_\phi-\tau)}.$$  (G.20)

Now using eq.(G.15) in the above equation, we obtain,

$$\Re(s) \leq \mathfrak{a}^{(p)}_{n,\Delta_\phi/2} \frac{s^{n+1}}{s-\frac{\Delta_\phi}{2}} \widehat{Q}_{L+2}\left(s+\frac{\Delta_\phi}{2}\right).$$  (G.21)

Next we will analyze $\widehat{Q}_{L+2}\left(s+\Delta_\phi/2\right)$. We will analyze this in the limit of large $L$ first. For this purpose we will look into large $L$ asymptotic of the $\widehat{Q}_{L+2}(s+\Delta_\phi/2)$. This asymptotic was worked out in [46] and is given by the equation (A.23) therein. Using the formula we have,

$$\widehat{Q}_{L+2}(s+\Delta_\phi/2) \sim (s)^2_{\frac{\Delta_\phi}{2}} \left[\left(L+2+s+\frac{\Delta_\phi}{2}\right)\left(L+1+s+\frac{\Delta_\phi}{2}\right)\right]^{-\frac{\Delta_\phi}{2}}.$$  (G.22)

Thus can write asymptotically,

$$\Re(s) \leq \mathfrak{a}^{(p)}_{n,\Delta_\phi/2} s^{n+\Delta_\phi}(L+s)^{-\Delta_\phi}.$$  (G.23)

We can use this inequality to find the optimal value of $L$. The idea is that the remainder term is exponentially small. Explicitly, first we cast the RHS of the inequality above in the following form,

$$e^{\ln \mathfrak{a}^{(p)}_{n,\Delta_\phi/2}+(n+\Delta_\phi)\ln s-\Delta_\phi \ln(L+s)},$$  (G.24)

---

[18]This has been checked numerically on Mathematica.

which leads to

$$L = s\left[(s^n \mathfrak{a}_{n,\Delta_\phi/2}^{(p)})^{\frac{1}{\Delta_\phi}} - 1\right].$$

(G.25)

We note that if $\Delta_\phi \gg 1$ then we have essentially the leading asymptotic for $L$,

$$\boxed{L \approx \frac{n}{\Delta_\phi} s \ln s}.$$

(G.26)

Interestingly, this $s \ln s$ behavior was also found in [25] giving rise to the so called Greenberg-Low bound, which is weaker than the Froissart bound.

## G.2 Summing over twists

With this, next we move on to bounding $\bar{\mathcal{A}}_M^{(p)}(s)$. Now if we assume that $L$ is such that in the large $s$ limit the remainder term $\mathfrak{R}(s)$ is vanishingly small then we can effectively cut the $\ell$ sum at $\ell = L$. Thus we have,

$$\bar{\mathcal{A}}_M^{(p)}(s) = \frac{\pi}{s - \frac{\Delta_\phi}{2}} \sum_{\substack{\ell=0 \\ \ell \text{ even}}}^{L} \sum_{\tau=\Delta_\phi}^{s+\frac{\Delta_\phi}{2}} C_{\tau,\ell} \frac{\Gamma(2\tau+2\ell)}{\Gamma^2(\tau)\Gamma^2(\tau+\ell)\Gamma^2(\Delta_\phi-\tau)} \widehat{Q}_\ell(\tau).$$

(G.27)

Next using the fact $\widehat{Q}_\ell(\tau) \le 1$ we can write,

$$\bar{\mathcal{A}}_M^{(p)}(s) \le \frac{2\pi}{2s - \Delta_\phi} \sum_{\substack{\ell=0 \\ \ell \text{ even}}}^{L} \sum_{\tau=\Delta_\phi}^{s+\frac{\Delta_\phi}{2}} C_{\tau,\ell} \frac{\Gamma(2\tau+2\ell)}{\Gamma^2(\tau)\Gamma^2(\tau+\ell)\Gamma^2(\Delta_\phi-\tau)}.$$

(G.28)

We follow the same strategy as the one in the main text. What we will do is to put for the conformal block coefficient its MFT value

$$C_{\tau,\ell}^{MFT} = \frac{2\Gamma(\ell+h)\Gamma^2(\ell+\tau)\Gamma(\ell+2\tau-1)\Gamma^2(-h+\tau+1)}{\Gamma(\ell+1)\Gamma(2\ell+2\tau-1)\Gamma(-2h+2\tau+1)\Gamma(\ell-h+2\tau)} \\ \times \frac{\Gamma(-2h+\tau+\Delta_\phi+1)\Gamma(\ell-h+\tau+\Delta_\phi)}{\Gamma^2(\Delta_\phi)\Gamma(\tau-\Delta_\phi+1)\Gamma^2(-h+\Delta_\phi+1)\Gamma(\ell+h+\tau-\Delta_\phi)}$$

(G.29)

and do the analysis. In the limit $\tau \gg \ell$ while also considering $\tau \gg 1$ the $\tau$ summand asymptotes to

$$C_{\tau,\ell}^{MFT} \frac{\Gamma(2\tau+2\ell)}{\Gamma^2(\tau)\Gamma^2(\tau+\ell)\Gamma^2(\Delta_\phi-\tau)} \sim \frac{2^{3h-2\tau+1}\Gamma(\ell+h)\tau^{2\Delta_\phi-3h+\frac{5}{2}}}{\pi^{3/2}\Gamma(\ell+1)\Gamma(\Delta_\phi)^2\Gamma^2(-h+\Delta_\phi+1)} \sin^2\left[\pi(\Delta_\phi-\tau)\right].$$

(G.30)

Now as before we will replace the sum over twist by an integral so that basically we are left with,

$$\int_{\Delta_\phi}^{s+\frac{\Delta_\phi}{2}} d\tau \, e^{-(\ln 4)\tau} \tau^{2\Delta_\phi-3h+\frac{5}{2}} \sin^2\left[\pi(\tau-\Delta_\phi)\right] \le \int_{\Delta_\phi}^{s+\frac{\Delta_\phi}{2}} d\tau \, e^{-(\ln 4)\tau} \tau^{2\Delta_\phi-3h+\frac{5}{2}},$$

(G.31)

where we have used $\sin^2\left[\pi(\tau-\Delta_\phi)\right] \le 1$ in the last step. Now considering $s \gg \Delta_\phi$ we introduce the rescaled variable $\hat{\tau}$ defined by

$$\hat{\tau} := \frac{\tau}{s}.$$

(G.32)

In terms of this variable the integral above translates into,

$$s^{\frac{7}{2}+2\Delta_\phi-3h} \int_0^1 d\hat{\tau}\; e^{-s\ln 4\hat{\tau}}\; \hat{\tau}^{\frac{5}{2}+2\Delta_\phi-3h}.$$ (G.33)

Now the large $s$ asymptotic of the integral is,

$$s^{\frac{7}{2}+2\Delta_\phi-3h} \int_0^1 d\hat{\tau}\; e^{-s\ln 4\hat{\tau}}\; \hat{\tau}^{\frac{5}{2}+2\Delta_\phi-3h} \sim (\log 4)^{-2\Delta_\phi+3h-\frac{7}{2}}\Gamma\left(-3h+2\Delta_\phi+\frac{7}{2}\right) - \frac{4^{-s}s^{2\Delta_\phi-3h+\frac{5}{2}}}{\log(4)}.$$ (G.34)

Now clearly the first term dominates over the second above in the limit $s \gg \Delta_\phi \gg 1$ so that we can finally use

$$\sum_{\tau=\Delta_\phi}^{s+\frac{\Delta_\phi}{2}} C_{\tau,\ell}\frac{\Gamma(2\tau+2\ell)}{\Gamma^2(\tau)\Gamma^2(\tau+\ell)\Gamma^2(\Delta_\phi-\tau)} \sim (\log 4)^{-2\Delta_\phi+3h-\frac{7}{2}}\Gamma\left(-3h+2\Delta_\phi+\frac{7}{2}\right).$$ (G.35)

## G.3 Finally the bound

Putting this crucial piece of information into eq.(G.28) we obtain,

$$\bar{\mathcal{A}}_M^{(p)} \le \frac{\pi^{-1}}{2s-\Delta_\phi}(\ln 4)^{3h-2\Delta_\phi-\frac{7}{2}}\,\Gamma\left(2\Delta_\phi+\frac{7}{2}-3h\right)\frac{2^{2h-3}\sqrt{\pi}}{\Gamma^2(\Delta_\phi)\Gamma^2(1-h+\Delta_\phi)}\sum_{\substack{\ell=0\\ \ell\text{ even}}}^{L}(\ell+1)_{h-1}.$$ (G.36)

Now we do the sum,

$$\sum_{\substack{\ell=0\\ \ell\text{ even}}}^{L}(\ell+1)_{h-1} = \frac{(L+2)\Gamma\left(\frac{2h+L+2}{2}\right)}{2h\Gamma\left(\frac{L+4}{2}\right)}.$$ (G.37)

Now since in general $L$ is large hence we can consider the large $L$ asymptotic of the above sum and thus we can write to the leading order in the large $L$ asymptotic,

$$\sum_{\substack{\ell=0\\ \ell\text{ even}}}^{L}(\ell+1)_{h-1} \sim \frac{2^{-h}L^h}{h}.$$ (G.38)

Note that we have made extensive use of the assumption $L \ll s$ in the previous section to reach upto this point. As explained before this is possible when $\Delta_\phi \gg 1$. Thus we will now put this constraint into its place. Thus using eq.(G.26) and considering the limit $s \gg \Delta_\phi/2$ one has the following asymptotic bound on $\bar{\mathcal{A}}_M^{(p)}$,

$$\bar{\mathcal{A}}_M^{(p)} \le \mathcal{A}_0\, s^{h-1}\ln^h s\,,$$ (G.39)

with

$$\mathcal{A}_0 = (\ln 4)^{-2\Delta_\phi}\,\Gamma\left(-3h+2\Delta_\phi+\frac{7}{2}\right)\frac{2^{h-4}\pi^{-1/2}h^{-1}}{\Gamma^2(\Delta_\phi)\Gamma^2(1-h+\Delta_\phi)}\left(\frac{n}{\Delta_\phi}\right)^h.$$ (G.40)

Now at this point we would like to comment on the main purpose of this exercise. To do so we compare $\mathcal{A}_0$ above with $\mathcal{B}_1, \mathcal{B}_2, \mathcal{B}_3$ from eq.(4.46), eq.(4.50), eq.(4.55) respectively. In each case we observe that $\mathcal{A}_0$ is exponentially suppressed in the limit $\Delta_\phi \to \infty$ i.e., the flat space limit under consideration. Thus, this matches with our expectation as described at the beginning of this appendix.

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
