# Peer review of "Froissart bound for/from CFT Mellin amplitudes"

_SciPost Physics, doi:SciPost Phys. 8, 095 (2020)_

## Round 1 · Referee Report · Connor Behan · 2020-5-3

Strengths

1. Interesting result with connections to collider experiments.
2. Starts off with two helpful overviews, one for each side of the correspondence.
3. Excellent use of appendices to make section 4 flow nicely.
4. Provides a great demonstration of the progress that can be gained by combining old and new work on the S-matrix.

Weaknesses

1. There are a few typos discussed below.
2. "Flat-space" vs "flat space", $\Gamma^2(x)$ vs $\Gamma(x)^2$ and $u$ vs $\hat{u}$ are not consistent.
3. Some of the limits being used are more rigorous than others.

Report

This article presents a new method for bounding the absorptive part of a massive scattering amplitude. Instead of working in flat space, the authors take the background to be a nearly flat AdS space which allows them to exploit the duality with a CT (nonlocal CFT) in one lower dimension. This alone would already make for an interesting paper. However, the final result contains some surprises compared to the usual Froissart bound. Specifically, it is stronger for (bulk) dimension d < 4. Even for d = 4, when the bound has the same form, the authors find the remarkable fact that the exchanged mass can be replaced with the external mass yielding a better fit to experimental data. Apparently, after assuming polynomial boundedness in s for some ellipse, t can be deformed to a more forgiving value in this formulation.

Since the paper is highly technical, there are a few steps that could benefit from more explanation. First, why does the $\sin^2$ disappear after (4.14)? Second, the discussion around (4.3) says $\Delta_\phi \gg 1, \tau \gg 1$ but it should also say they are of the same order. Was the sum (4.19), which supports this relation between $\Delta_\phi$ and $\tau$, found by assuming it in the first place? If so, there should be some argument about why it is safe to take this limit self-consistently. Third, it would be nice to read more about which part of the outer sum dominates in (4.37). It starts from zero but various assumptions about the spin being large are used inside the sum.

My other requests are just about typos. SciPost should look forward to publishing the final version.

Requested changes

1. Update the reference to appendix E on page 8 so that is says appendix G.
2. Footnote 5 should probably say "which will make a difference".
3. Appendix A says "for" instead of "form" and appendix D says "This" instead of "this".
4. The line under (E.1) should say "able to approximate this sum" or something similar.
5. Sentences that would sound better with articles are "there exists A certain" on page 5 and "obtaining AN asymptotic upper bound" on page 15.
6. The sentence under (3.2) should say there are $n(n-3)/2$ independent variables because the specialization to 4 points is only made in the next sentence.
7. There appears to be a missing division sign in (B.4), a $45m^2$ in (C.9) and unmatched parentheses in (D.7).
8. It is not obvious that the flat space limit would lead to $\tau \sim \Delta_\phi$ and $\ell \gg 1$. If it is possible to eliminate all doubts about this, that should be done in section 4.1.

---

## Round 1 · Referee Report · Matthew Dodelson · 2020-5-6

Report

This paper initiates the program of generalizing the Froissart bound to holographic conformal field theories in the flat space limit. The Mellin amplitude in conformal field theory plays the role of the scattering amplitude in ordinary quantum field theory, and the bounds derived in this paper are on the imaginary part of the Mellin amplitude. Just like in the flat space case, the authors use the assumption of polynomial boundedness to derive their bounds. Both the forward and nonforward limits are discussed.

This is an important and original paper, and I strongly suggest its publication.

Requested changes

1. The authors state that the flat space limit involves taking R/l_planck large. I believe that one also requires the dimensionless quantity R/l_string to be large.

2. In equation (3.26), the upper bound of the integral at infinity is omitted.

3. Some minor spelling errors: on page 3, unitarty should be unitarity. On page 4, analytitcity should be analyticity. On page 5, indepnpendent should be independent. On page 7, prcatical should be practical. On page 13, puting should be putting. On page 15, depepndency should be dependency. On page 20, hyperegometric should be hypergeometric.

---

## Round 5 · Referee Report · Anonymous (Referee 3) · 2020-6-13

Report

The paper looks ready to be published now. The recent changes are much appreciated.

---

## Round 5 · Referee Report · Matthew Dodelson (Referee 2) · 2020-6-15

Report

The authors have completed the suggested edits, so I would recommend that the article be published.

---

## Round 5 · List of Changes

Clarifications Added Various clarifications and comments are added in accordance with the referee reports. These are listed below. 1. Following equation (3.11), it has been mentioned that the flat space limit we are also considering is where $R/\ell_{strings}\gg 1$. This is because in the flat space limit we are considering a massive quantum field theory. 2. In the sentence under (3.2), we have added that the general Mellin amplitude for $n-$point conformal correlator depends upon $n(n-3)/2$ independent variables. 3. In section 4.1, various footnotes have been added to add clarifications regarding various assumptions and computational steps. These are delineated in footnotes 5-9.

Typos Fixed 1. In equation (3.26) upper bound of the integral at infinity is fixed. 2. Consistently "flat space" has been used instead of "Flat-space". 3. Notational inconsistency between $\Gamma^2(x)$ and $\Gamma(x)^2$ has been rectified to stick with $\Gamma^2(x)$. 4. On page 8 penultimate line appendix number has been corrected to G from E. 5. In (B.4) the missing division sign is restored, in (C.9) the $45m^2$ is corrected to $4m^2$ and the unmatched parenthesis in (D.7) is taken care of. 6. Spellings corrected at various places and the grammar is improved at places as par the referee recommendation.

---

## Editorial Decision

published